# Platelet-derived exerkine CXCL4/platelet factor 4 rejuvenates hippocampal neurogenesis and restores cognitive function in aged mice

Odette Leiter [1], David Brici[1], Stephen J. Fletcher[2], Xuan Ling Hilary Yong [1], Jocelyn Widagdo[1], Nicholas Matigian[3], Adam B. Schroer [4], Gregor Bieri [4], Daniel G. Blackmore[1], Perry F. Bartlett[1], Victor Anggono [1], Saul A. Villeda [4,5,6] & Tara L. Walker [1] ✉

The beneficial effects of physical activity on brain ageing are well recognised, with exerkines, factors that are secreted into the circulation in response to exercise, emerging as likely mediators of this response. However, the source and identity of these exerkines remain unclear. Here we provide evidence that an anti-geronic exerkine is secreted by platelets. We show that platelets are activated by exercise and are required for the exercise-induced increase in hippocampal precursor cell proliferation in aged mice. We also demonstrate that increasing the systemic levels of the platelet-derived exerkine CXCL4/platelet factor 4 (PF4) ameliorates age-related regenerative and cognitive impairments in a hippocampal neurogenesis-dependent manner. Together these findings highlight the role of platelets in mediating the rejuvenating effects of exercise during physiological brain ageing.

There is overwhelming evidence that regular physical exercise can counteract cognitive decline in both healthy ageing and in neurodegenerative conditions such as Alzheimer's disease[1–3]. However, for many people the prospect of exercising is not feasible due to health conditions, mobility limitations or advanced age. There is therefore a burgeoning scientific and commercial interest in the identification of pharmacological interventions that can mimic or potentiate the effects of exercise—so called "exercise pills" or "exercise mimetics". However, the molecular mechanisms underlying the beneficial effect of exercise on brain health remain poorly understood. In animal models, exercise promotes adult neurogenesis in the hippocampus[4], and mediates associated spatial learning and memory function[5], with exerkines, the

circulating humoral factors released into the blood stream in response to exercise, emerging as likely mediators of this effect[6–8]. It has recently been shown that the beneficial effects of exercise can be transferred to the hippocampus of aged mice through administration of blood plasma from exercising mice, thereby promoting neurogenesis and enhancing cognition[6]. Several tissue sources of exerkines have been suggested, including the liver[6,7] and muscle[8]. However, we have recently demonstrated an unexpected role of platelets, the small anucleate immune cells which are primarily known for their role in haemostasis, in mediating the exercise-induced increase in adult hippocampal neurogenesis. We found that platelets are activated by acute exercise and release humoral factors including the chemokine platelet

[1]Clem Jones Centre for Ageing Dementia Research, Queensland Brain Institute, The University of Queensland, Brisbane, QLD 4072, Australia. [2]Centre for Horticultural Science, Queensland Alliance for Agriculture and Food Innovation, The University of Queensland, Brisbane, QLD 4072, Australia. [3]Queensland Cyber Infrastructure Foundation Ltd, The University of Queensland, Brisbane, QLD 4072, Australia. [4]Department of Anatomy, University of California San Francisco, San Francisco, CA 94143, USA. [5]The Eli and Edythe Broad Center of Regeneration Medicine and Stem Cell Research, University of California San Francisco, San Francisco, CA 94143, USA. [6]Bakar Aging Research Institute, University of California San Francisco, San Francisco, CA 94143, USA. ✉e-mail: t.walker1@uq.edu.au

factor 4 (PF4)[9]. We also showed that PF4 is sufficient to enhance hippocampal neurogenesis when delivered directly to the brain of young mice via osmotic pumps[9]. However, whether platelet-derived exerkines such as PF4 can recapitulate the rejuvenating effects of exercise on neurogenesis and rescue cognitive decline in ageing when delivered systemically remains unknown.

Here, we identify platelets as necessary mediators of the rejuvenating effects of exercise in aged mice. We show that the platelet-released exerkine PF4 is both sufficient and necessary to mediate this process. Furthermore, we demonstrate that systemic PF4 delivery can mimic the rejuvenating effects of exercise by enhancing adult hippocampal neurogenesis and restoring cognitive function in the aged brain. Using an aged transgenic mouse model of neurogenesis ablation, we also show that neurogenesis is necessary for PF4-mediated cognitive rejuvenation.

## Results

### Systemic PF4 promotes adult neurogenesis and is necessary for its exercise-induced increase

Given that the exerkine PF4 can enhance adult hippocampal neurogenesis in young mice following central administration[9], we first investigated whether peripherally delivered PF4 promotes hippocampal neurogenesis. To do this we mimicked the increase in plasma PF4 levels following exercise by injecting PF4 (500 ng) into the tail vein of young adult mice every second day for 1 week (Fig. 1a). Similar to the effect observed following systemic delivery of plasma isolated from

exercised aged mice[6], systemically delivered PF4 increased the number of doublecortin[+] (DCX[+]) immature neurons in the dentate gyrus without affecting neural precursor cell proliferation (Ki67[+] cells) in young adult mice (Fig. 1b, c). Interestingly, this contrasts with the effect of exercise on adult neurogenesis, which we have recently shown increases the number of proliferating neural precursor cells, likely via the recruitment of quiescent neural stem cells into the neurogenic trajectory[10]. To confirm that systemic PF4, like plasma isolated from exercising mice, does not affect the recruitment of quiescent stem cells, we used a dual thymidine analogue labelling paradigm (using 5-chloro-2'-deoxyuridine (CldU) and 5-iodo-2'-deoxyuridine (IdU)) and a single systemic injection of PF4 (Fig. 1d). We observed no difference in the proportion of CldU[−]IdU[+] cells in response to acute PF4 treatment, indicating that previously quiescent cells were not being recruited into the proliferative cycle (Fig. 1e). We also saw no change in the percentage of CldU[+] cells with or without IdU label, suggesting that the PF4 treatment did not affect cells that were proliferative prior to treatment (Fig. 1e). Together these findings suggest that an increase in the level of PF4 in the plasma of young mice does not cause a proliferative response, including the recruitment of neural stem cells from quiescence, but instead enhances the neurogenic process at later stages, such as through the survival or maturation of newborn neurons.

To further investigate the effects of PF4 on neural precursor cell dynamics, we next turned to the in vitro adherent hippocampal monolayer culture model[11,12]. In this model, proliferating adult neural

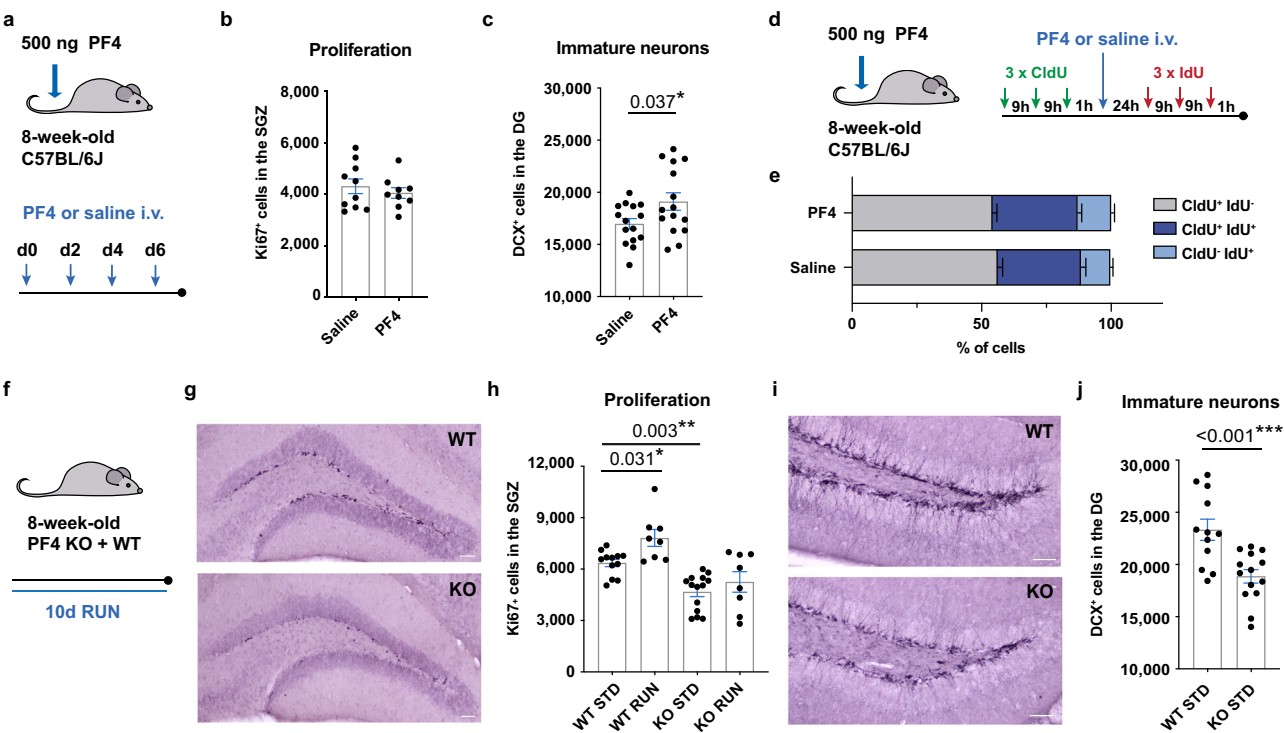

**Fig. 1 | Systemic PF4 enhances adult hippocampal neurogenesis in vivo.**
**a** Experimental design of PF4 injection paradigm in young mice. **b, c** Intravenous (i.v.) PF4 injections for 1 week did not affect neural precursor cell proliferation in the subgranular zone (SGZ; *n* = 10 mice in saline group; *n* = 9 mice in PF4 group; counts of one hemisphere), but increased the number of doublecortin[+] (DCX[+]) cells (*n* = 15 mice per group; counts of one hemisphere). **d** Experimental design of the double-labelling paradigm with CldU and IdU. **e** Acute administration of PF4 did not affect neural precursor cell proliferation, including the recruitment of cells from quiescence (*n* = 20 mice per group). **f** Running paradigm of PF4 knockout (KO) mice. **g** Representative images of Ki67[+] cells in the SGZ of PF4 KO and wildtype (WT) mice. Scale bar: 50 μm. **h** PF4 KO mice show a significant reduction in the number of

proliferating cells compared to wildtype mice. Physical exercise did not increase neural precursor proliferation in PF4 KO mice (WT STD *n* = 12; WT RUN *n* = 8; KO STD *n* = 14; KO RUN *n* = 8; counts of one hemisphere). **i** Representative images of DCX[+] cells in the dentate gyrus (DG) of PF4 KO and wildtype mice. Scale bar: 50 μm. **j** PF4 KO mice have significantly lower levels of baseline neurogenesis compared to wildtype littermates (WT STD *n* = 12; KO STD *n* = 14; counts of one hemisphere). STD standard-housing, RUN 10-day running. Bars are mean ± SEM. Statistical analysis was performed using unpaired Student's two-tailed *t* tests in (**c**) and (**j**), and one-way ANOVA with Sidak comparison in (**h**). **p* < 0.05, ***p* < 0.01, ****p* < 0.001. Source data are provided as Source Data file.

precursor cells are cultured as a homogenous population under controlled conditions, allowing monitoring of their cellular changes in response to PF4 treatment, including potential changes in proliferation, differentiation and morphology[13]. To first determine whether PF4 is taken up directly by adult neural precursor cells, we treated the cells with recombinant mouse PF4 protein conjugated to Alexa Fluor 568 for 2, 6 and 24 h. Using confocal microscopy, we detected labelled PF4 within the cells following 2 h of incubation, with an increase in fluorescent signal after 6 h, remaining at a similar intensity for at least 24 h (Supplementary Fig. 1a, b), indicating that PF4 is taken up and internalised by the adult neural stem cells. Our in vivo data suggested that PF4 confers a pro-survival or differentiation effect on the neural precursor cells rather than affecting their proliferative capacity. To confirm this finding, we next performed an assay in which proliferating adherent hippocampal precursor cells were grown in the presence of PF4 and labelled with the thymidine analogue bromodeoxyuridine (BrdU). In accordance with the data from our in vivo experiments, we observed no difference in the number of BrdU-labelled cells between PF4-supplemented and control cultures, suggesting that PF4 had no effect on the proliferative capacity of these cells (Supplementary Fig. 1c, d). In support of this we also found no difference in the proportion of PF4-treated cells in the S and G2/M phases of the cell cycle (Supplementary Fig. 1e, f). However, we did observe that a higher proportion of cells entered the G1/G0 phases after PF4 treatment (Supplementary Fig. 1f), indicating that they had left their proliferative phase and started to differentiate into neurons. To confirm whether PF4 could stimulate neuronal differentiation, we performed a differentiation assay on the adherent cultures and quantified the number of cells that had become glial fibrillary acidic protein (GFAP)+ astrocytes and β-III-tubulin+ neurons (Supplementary Fig. 1g, h). In support of our previous data generated from differentiated neurosphere cultures[9], we found a significant increase in β-III-tubulin+ cells in the PF4-treated adherent cultures and no effect on astrocyte differentiation. Together these findings highlight the pro-neurogenic effect of PF4.

Having identified a robust pro-neurogenic effect of PF4 treatment in vitro and in vivo, we next investigated whether PF4 is required for the maintenance of baseline levels of adult hippocampal neurogenesis. For this analysis we used 8-week-old PF4 knockout mice, in which platelets are devoid of PF4 mRNA or protein[14]. Although these mice exhibit a higher platelet count compared to wildtype littermates, other haematologic parameters, such as platelet function, as well as their overall appearance and anatomy, weight, survival and fertility, remain unaffected[14]. We found that the levels of baseline proliferation and neurogenesis were significantly lower in the dentate gyrus of mice lacking PF4, as evidenced by fewer proliferating Ki67+ cells and DCX+ immature neurons (Fig. 1f–j). This reduction was not due to changes in the volume of the hippocampus or the granular cell layer, as no differences between wildtype and knockout littermates were observed (Supplementary Fig. 2a–c). We also found that PF4 deletion did not affect the proliferation of neural precursor cells in the subventricular zone, the other major neurogenic niche in the adult mammalian brain (Supplementary Fig. 2d, e), suggesting a specific effect of PF4 on adult hippocampal neurogenesis. This is in line with our previous observation, in which PF4 treatment specifically promoted neurogenesis in neurosphere cultures derived from the dentate gyrus but had no effect on subventricular zone-derived cultures[9]. Given that exercise strongly increases neural precursor cell proliferation in the dentate gyrus, but not the subventricular zone[15], we next sought to determine whether PF4 is also required for the proliferative response of neural precursor cells to exercise. To examine this, we housed a subset of PF4 knockout and wildtype mice in cages with running wheels for 10 days. As expected, exercise robustly increased the number of proliferating neural precursor cells in the hippocampus of wildtype mice (Fig. 1f–h). However, although wildtype and knockout mice ran comparable distances (Supplementary Fig. 2f, g), the exercise-induced increase in

precursor cell proliferation was absent in mice lacking PF4, suggesting that it is required for this response (Fig. 1h).

To gain further insight into the mechanism by which PF4 affects adult hippocampal neural precursor cells, we performed ribonucleic acid (RNA) sequencing. To do this, adult primary dentate gyrus-derived neural precursors were treated with either PF4 or saline, then isolated by flow cytometry using a biotinylated epidermal growth factor (EGF) complexed to a fluorescent marker (Fig. 2a and Supplementary Fig. 3a). RNA sequencing of six samples from each cell population revealed several changes in the transcriptomic signature of EGF+ adult neural precursor cells 2 h after PF4 treatment, including 270 genes that were significantly upregulated and 386 genes which were significantly downregulated (Fig. 2b, c, Supplementary Fig. 3c and Supplementary Data 1). PF4 induced fewer changes in other dentate gyrus cells (EGF− cell population), in which our analysis revealed 110 differentially expressed genes (73 genes upregulated and 37 downregulated; Fig. 2b, Supplementary Fig. 3d, e and Supplementary Data 2). A comparison of the significantly upregulated and significantly downregulated genes in both cell populations (EGF+ and EGF− cells) revealed only 9 and 5 overlapping genes, respectively (Fig. 2d), suggesting differential effects of PF4 treatment in different cell populations. We next performed a gene ontology (GO) enrichment analysis of the differentially expressed genes induced by PF4 in the EGF+ neural precursor cell population. This revealed 96 biological processes that were significantly enriched ($p < 0.05$) amongst the upregulated genes and 39 biological processes amongst the genes that were downregulated following PF4 treatment (Fig. 2e and Supplementary Data 3). In accordance with our data showing a pro-neurogenic effect of PF4, the upregulated genes revealed an enrichment in several GO categories involved in neuronal differentiation, including "cell differentiation" (GO:0030154; $p = 2.1 \times 10^{-5}$), "generation of neurons" (GO:0048699; $p = 2.8 \times 10^{-4}$) "neuron differentiation" (GO:0030182; $p = 0.002$) and "neuron projection development" (GO:0031175; $p = 0.01$) (Supplementary Data 3). When viewed as a STRING protein-protein interaction network[16], many of the genes which showed enrichment in GO biological processes related to differentiation also clustered together, suggesting a functional relevance and similarity of these genes (Fig. 2f). Notably, Markov clustering of the genes that were upregulated in EGF+ cells following PF4 supplementation also revealed a small STRING network of genes which showed functional enrichment in the GO categories "long-term memory" (GO:0007616; $p = 0.005$), "learning or memory" (GO:0007611, $p = 0.005$) and "positive regulation of synaptic transmission" (GO:0050806, $p = 0.007$), suggesting that PF4 increases the expression of genes associated with learning and memory and synaptic transmission in adult neural precursor cells (Fig. 2g). A GO enrichment analysis of genes that were changed in the EGF− population returned no results, suggesting that PF4 has specific effects on proliferative EGF+ adult neural precursor cells.

In addition to its beneficial effects on adult neural precursor cells, exercise also promotes synaptic plasticity in dentate granule neurons by enhancing dendritic complexity and length as well as increasing the density of dendritic spines[17]. To test whether PF4 induces synaptic changes in mature neurons, we performed a western blot analysis on dentate gyrus tissue isolated from 8-week-old C57BL/6J mice that had received systemic PF4 injections (500 ng) every second day for 1 week. Analysis of the protein levels of PSD-95, a postsynaptic marker, and synaptophysin, a presynaptic marker, revealed no changes in the crude synaptosomal fraction (P2) or in the total homogenate of dentate gyrus tissue following PF4 treatment (Supplementary Fig. 4a–c), suggesting that PF4 does not affect the synaptic composition of mature neurons. To confirm this, we analysed the neurite complexity of primary hippocampal neurons treated with 100 ng/ml PF4, a dose we used previously on primary neurosphere cultures and in our adherent monolayer studies of neural precursor cells. A Sholl analysis revealed no effect of PF4 on the neurite complexity of mature neurons

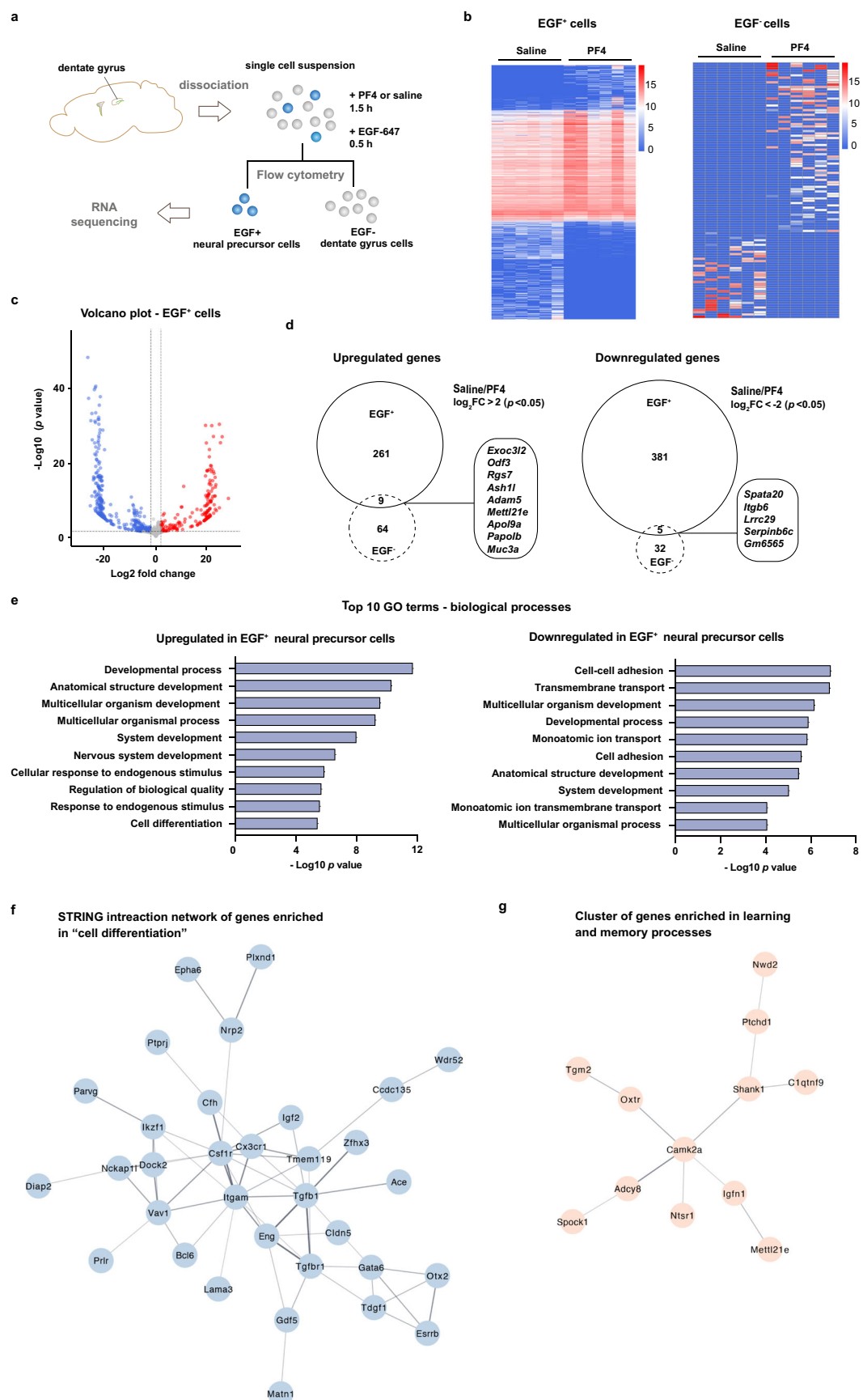

**Fig. 2 | PF4 induces transcriptomic changes in adult neural precursor cells.**
**a** Schematic illustration. Primary cells were isolated from the dentate gyrus of 8-week-old C57BL/6J mice and treated with either 100 ng/ml PF4 or saline, then stained with epidermal growth factor (EGF) conjugated with Alexa Fluor® 647 (EGF-647). EGF⁺ and EGF⁻ cells were collected using flow cytometry followed by RNA sequencing of six samples from each cell population. **b** Heatmaps of significantly changed genes induced by PF4 in EGF⁺ cells (left) and EGF⁻ cells (right). Colour code represents log₂ normalised expression. **c** Volcano plot of differentially expressed genes in EGF⁺ cells treated with PF4 or saline, with significantly upregulated genes in red (log₂ fold change >2 and adjusted *p* value < 0.05) and significantly downregulated genes in blue (log₂ fold change < −2 and adjusted *p* value < 0.05). Wald test and Benjamini–Hochberg correction. **d** Euler diagrams comparing upregulated (left side) and downregulated (right side) genes between EGF⁺ cells and EGF⁻ cells

following 24 h of culture, suggesting that PF4 does not alter the morphology of these cells (Supplementary Fig. 4d, e).

### Exercise alters the platelet proteome in young and aged mice

Having identified PF4 as a platelet-derived exerkine that is sufficient and necessary for the regulation of adult neurogenesis in young mice, we next sought to gain further insight into the molecular changes that occur in platelets following 4 days of exercise and whether these are conserved in aged animals. Using a mass spectrometry-based proteomic analysis of platelets isolated from young and aged standard-housed and running mice, we identified 183 proteins that were significantly changed in the platelet lysate of young running mice (123 upregulated and 60 downregulated; Fig. 3a, b and Supplementary Data 4) and 164 proteins in aged running animals (143 upregulated and 21 downregulated; Fig. 3b and Supplementary Data 5). Although the protein changes differed between young and aged animals, we identified five proteins that were significantly upregulated at both ages following running (Fig. 3c): progranulin, properdin, tropomyosin, destrin and ubiquitin-associated and SH3 domain-containing protein B. Of these, progranulin, a glycosylated protein that has recently been shown to promote neurogenesis following injury[18], is particularly interesting, as raised progranulin levels in the hippocampus following exercise have previously been reported[19]. Although increased, the PF4 protein levels in platelets of young exercising mice (for 4 days) did not reach statistical significance when compared to standard-housed animals (2786 mean A.U. in standard-housed mice vs. 3969 mean A.U. in running mice; fold change: 1.42, *p* = 0.05, *n* = 5 mice per group). Given that platelet activation peaks in 8-week-old mice after 4 days of exercise[9], it is likely that the platelets have already released most of the PF4 content into the blood at this timepoint. This is consistent with our previous work showing a significant increase in plasma PF4 levels following short periods (1–4 days) of exercise in young mice[9]. We also identified several proteins with antioxidant capacity that were upregulated in the platelets of aged mice. These included selenoproteins and glutaredoxin-related protein 5, which is involved in redox control[20] (Supplementary Data 5).

To determine the biological processes that are involved in the platelet response to exercise, we performed a GO enrichment analysis of the proteins which were upregulated by exercise and found 393 biological processes that were significantly enriched (*p* < 0.05) amongst the proteins of young mice and 233 in those of aged mice (Fig. 3d and Supplementary Data 6 and 7). Of these, 61 pathways were enriched in both young and aged exercising animals (Fig. 3e). Among the common GO terms were processes highlighting the involvement of exercise-activated platelets in immune responses and molecule release mechanisms, as well as complement-related pathways. Moreover, proteins clustering in processes such as "positive regulation of neuron projection development" (GO:0010976; *p* = 0.02 in both young and aged mice) and "negative regulation of neuron apoptotic process" (GO:0043524; *p* = 0.04 in young mice and *p* = 0.03 in aged mice) suggest a role of exercise-induced platelet proteomic

(Wald test and Benjamini–Hochberg correction). **e** Gene ontology (GO) enrichment analysis of significantly upregulated (left panel) and significantly downregulated (right panel) genes in EGF⁺ cells treated with PF4 compared to saline-treated controls. Bar graphs show the top 10 significantly enriched GO terms for biological processes ranked by false discovery rate-adjusted *p* values (g:SCS multiple testing correction with significance threshold of 0.05). For a complete list of GO terms please see Supplementary Data 1. **f** A STRING interaction network of genes enriched in the GO category "cell differentiation" reveals a distinct gene cluster (genes with no connection are not depicted). **g** Markov clustering of the genes that were upregulated in EGF⁺ cells following PF4 treatment revealed a STRING network of 12 genes involved in learning and memory. FC fold change. Source data are provided as Source Data file.

changes in neuronal tissue. Notably, the GO biological processes of exercise-induced platelet protein changes in young mice also included specific neurogenesis-associated terms, such as "positive regulation of neurogenesis" (GO:0050769; *p* = 0.04), "negative regulation of neural precursor proliferation" (GO:2000178; *p* = 0.04) and several terms suggesting regulatory involvement in dendrite morphogenesis (Fig. 3f and Supplementary Data 6). Together, these data show that exercise induces molecular changes in the platelets of both young and aged mice and suggests that these cells can regulate exercise-induced neurogenesis.

### Platelets are required for the exercise-induced increase in neural precursor cell proliferation in aged mice

Having identified PF4 as a pro-neurogenic platelet-derived exerkine and observed molecular changes elicited by exercise in platelets from young and aged mice, we next asked whether platelets are necessary to translate the rejuvenating effects of exercise to the aged brain. Both the proteomic data above and previous reports[21] suggest that the running period required to promote a neurogenic response differs between young and old animals. To first determine the optimal period of running required to elicit a neurogenic response in aged mice we housed 18-month-old C57BL/6J mice in cages with or without a running wheel for 4, 10, 21 or 28 days. In contrast to young mice, in which acute running periods generate robust increases in neural precursor cell proliferation[22], we found no neurogenic responses following short (4 days) or intermediate (10 or 21 days) periods of exercise in the aged mice (Supplementary Fig. 5a–d), whereas 28 days of running significantly increased the number of both proliferating precursor cells and immature neurons (Fig. 4a–d). We also assessed the number of newly generated mature neurons using BrdU incorporation and the mature neuron marker NeuN. Our results revealed an increase in the number of BrdU⁺NeuN⁺ newborn neurons following 28 days of exercise (Fig. 4e, f), suggesting that this paradigm also stimulates neuronal survival in aged mice.

We next investigated whether, and at what timepoint, platelets are activated by exercise in aged animals to determine if this correlated with the optimal running period. We collected blood from exercising and standard-housed 18-month-old mice longitudinally for 35 days and measured the number and activation state of platelets. The proportion of activated platelets was determined using flow cytometry and the pan platelet marker CD61 in combination with CD62P (Supplementary Fig. 6a). CD62P is located within the alpha granule membrane of resting platelets; however, upon activation it is translocated via the secretory pathway to the plasma membrane surface, making it a proxy for platelet activation and activation-associated molecule release from alpha granules[23]. We found that 28 days of exercise significantly increased the number of CD62⁺ activated platelets (Fig. 4g, h and Supplementary Table 1), without affecting the total number of platelets (Supplementary Fig. 6c, d). Corticosterone measurements revealed no hallmarks of chronic stress in these animals (Supplementary Fig. 6b). To determine whether the platelet response

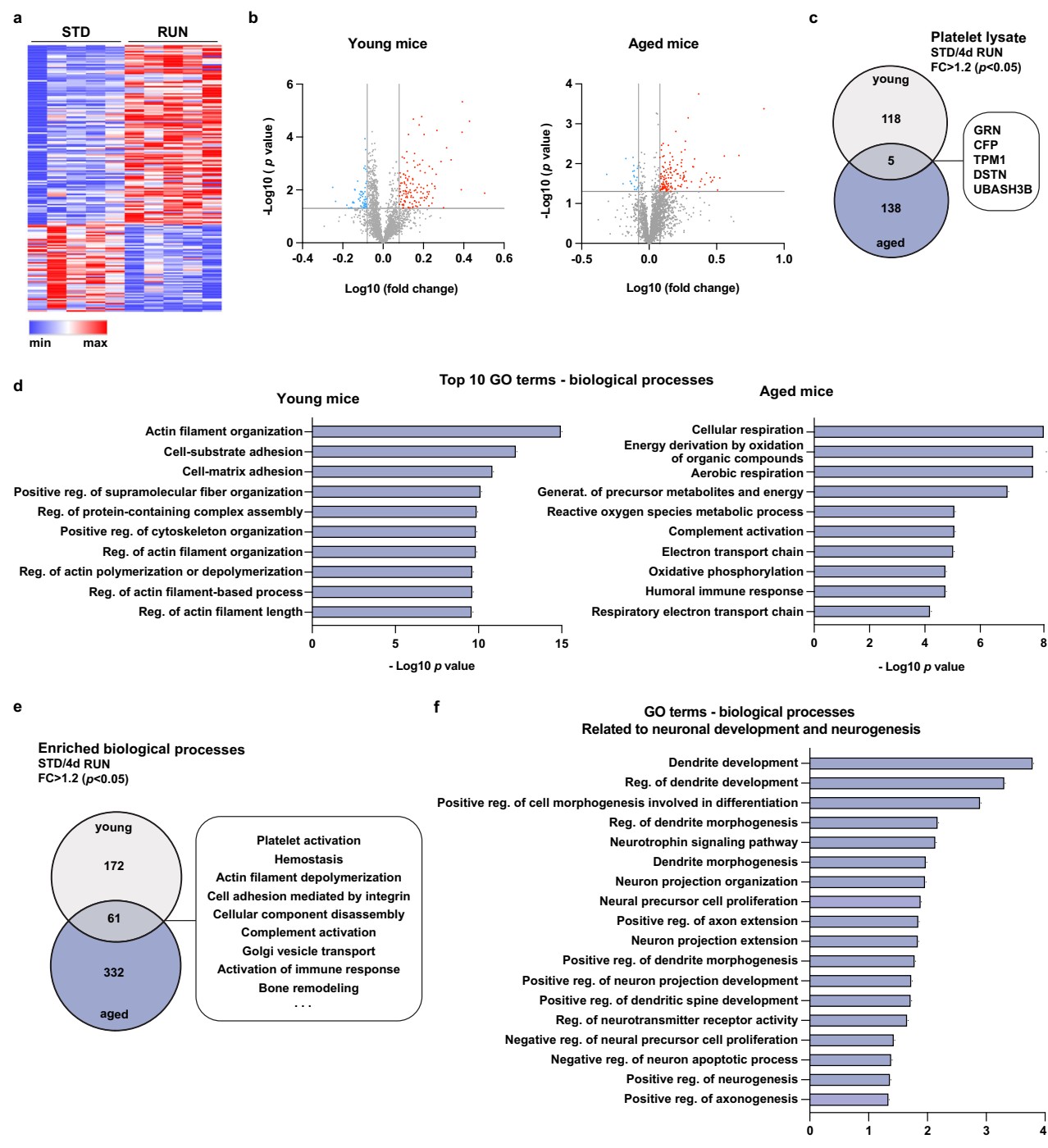

**Fig. 3 | Exercise alters the platelet proteomic signature in young and aged mice.** **a** Heatmap of platelet proteins significantly changed following 4 days of exercise in 8-week-old mice. **b** Volcano plots of quantified and differentially expressed proteins in the platelet lysate of 8-week-old (young) and 18-month-old (aged) mice following 4 days of running. Significantly upregulated proteins are labelled red ($p < 0.05$ and fold change >1.2) and significantly downregulated proteins are shown in blue ($p < 0.05$ and fold change < −1.2). Pairwise relative-abundance comparison using two-tailed *t*-tests. **c** Venn diagram showing the number of proteins significantly upregulated after 4 days of exercise in young and aged mice, including 5 proteins determined in both cohorts ($p < 0.05$ and fold change >1.2; pairwise relative-abundance comparison using two-tailed *t*-tests). **d** Gene ontology (GO) enrichment analysis of significantly increased proteins in the platelet lysate of young and aged running mice (for 4 days) compared to standard-housed controls. Bar graphs show the top 10 significantly enriched GO terms for biological processes

ranked by Benjamini–Hochberg false discovery rate-corrected *p* values. For a complete list of GO terms please see Supplementary Data 6 and 7. **e** Venn diagram showing the number of enriched biological processes following a GO analysis with proteins significantly increased after 4 days of exercise in young and aged mice ($p < 0.05$ and fold change >1.2; pairwise relative-abundance comparison using two-tailed *t*-tests). The box highlights some of the 61 biological processes that were enriched in both cohorts. **f** Neurogenesis- and neuronal development-related GO terms resulting from the GO enrichment analysis of significantly increased proteins in the platelet lysate of young running mice compared to standard-housed controls. Bar graphs show significantly enriched neurogenesis-related GO terms for biological processes ranked by Benjamini–Hochberg false discovery rate-corrected *p* values. STD standard-housing, RUN running, FC fold change. Source data are provided as Source Data file.

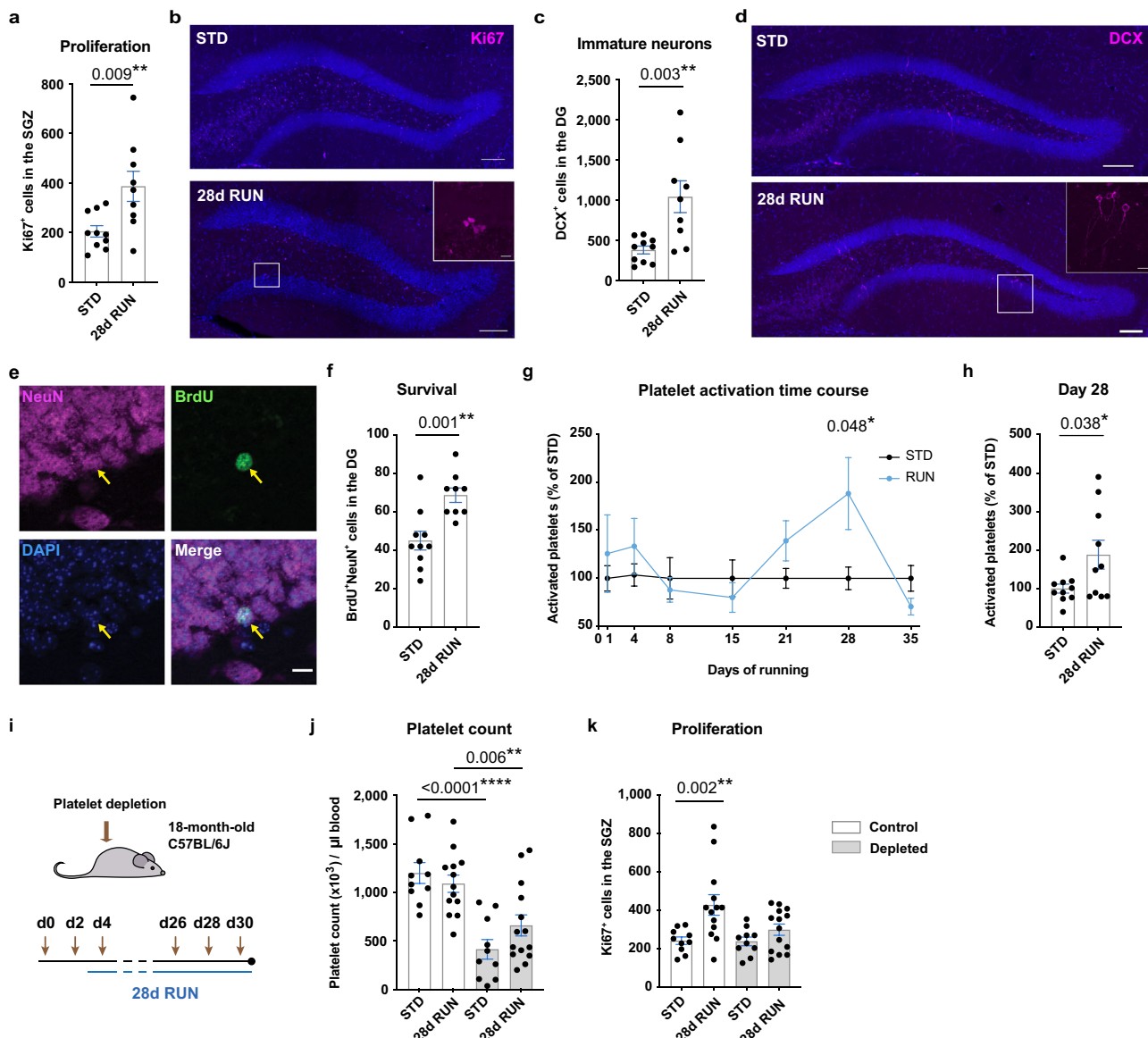

**Fig. 4 | Platelets are required for the exercise-induced increase in hippocampal neurogenesis in aged mice. a** Number of proliferating cells in the subgranular zone (SGZ) of 18-month-old C57BL/6J mice (STD $n = 10$ mice; 28 days RUN $n = 9$ mice). **b** Representative image of Ki67[+] cells in the dentate gyrus (DG). Scale bars: 100 and 10 μm in box highlighting a cluster of proliferating cells. **c** Number of DCX[+] cells in the DG of 18-month-old C57BL/6J mice (STD $n = 10$ mice; 28 days RUN $n = 9$ mice). **d** Representative image of DCX[+] cells in the DG. Scale bars: 100 μm and 15 μm in box highlighting a cluster of DCX[+] cells. **e** Representative image of BrdU/NeuN staining with yellow arrows indicating the same cell in each panel (BrdU−green; NeuN−magenta; DAPI−blue). Scale bar: 10 μm. **f** Number of BrdU[+]NeuN[+] cells in aged C57BL/6J mice (STD $n = 10$ mice; 28 days RUN $n = 9$ mice). **g** Time course of platelet activation showing a defined peak at 28 days of exercise ($n = 10$ mice per

group). **h** Individual values of platelet activation on day 28 ($n = 10$ mice per group). **i** Experimental design. **j** Platelet count in exercising and standard-housed mice receiving with control serum (white bars) or antiplatelet serum (grey bars; $n = 10$ mice in STD groups; $n = 13$ mice in 28 days RUN control group; $n = 14$ mice in 28 days RUN platelet-depleted group). **k** Number of proliferating cells in exercising and standard-housed mice receiving control serum (white bars) or antiplatelet serum (grey bars; $n = 10$ mice in STD groups; $n = 13$ mice in 28 days RUN control group; $n = 14$ mice in 28 days RUN platelet-depleted group). Bars are mean ± SEM. Statistical analysis was performed using unpaired Student's two-tailed $t$ tests in (**a**, **c**, **f**, **h**), two-way ANOVA with Sidak post hoc comparison in (**g**), and one-way ANOVA with Sidak post hoc comparison in (**j**) and (**k**). *$p < 0.05$, **$p < 0.01$, ****$p < 0.0001$. Source data are provided as Source Data file.

to longer-term exercise was similar to that observed following short bouts of exercise, we performed proteomic screening of platelets from aged mice following 28 days of running. Although we still identified 71 significantly changed proteins (55 upregulated and 16 downregulated; Supplementary Fig. 7 and Supplementary Data 5), GO analysis revealed that these changes were not directly associated with neurogenic responses, likely because neurogenesis-related proteins may have already been released into the blood at this running time-point. Together these data suggest that the platelet response to exercise coincides with the neurogenic response in aged animals.

To determine whether platelets are necessary for the running-induced increase in adult neurogenesis in aged mice, we next used antiplatelet serum to selectively deplete platelets[24] in 18-month-old mice, which were then housed (control or platelet depleted) in cages with or without a running wheel for 28 days (Fig. 4i). To ensure platelet depletion throughout the experiment, the test mice received antiplatelet serum every second day for 30 days, resulting in a significant decrease in platelet count (Fig. 4j). As expected, exercise led to a clear increase in the number of proliferating neural precursor cells in mice with a normal platelet count (Fig. 4k). However, although all the mice

spent a comparable time using the running wheels (Supplementary Fig. 8), the neurogenic response was absent in the aged animals after platelet depletion (Fig. 4k), thereby demonstrating that platelets are required for the running-induced increase in neural precursor proliferation. Together these results identify a previously unidentified regulatory role of platelets in exercise-induced adult neurogenesis in aged animals.

### The exerkine PF4 rejuvenates neurogenesis and cognitive function in aged mice

Age-related learning and memory deficits are associated with decreased adult hippocampal neurogenesis in mice but can be mitigated through physical exercise[25,26]. Having observed the necessity of platelets for the rejuvenating effects of exercise, we next asked whether the exerkine PF4 could mimic these effects in aged mice. In contrast to young mice, we found that raising the circulating PF4 plasma levels in 20-month-old C57BL/6J mice by administering PF4 (500 ng) via their tail vein every third day for 24 days significantly increased the number of proliferating cells in the subgranular zone of the dentate gyrus, compared to saline-treated animals (Fig. 5a–c). To determine which cell types accounted for this increase, we performed several histological analyses. We first tested whether oligodendrocyte progenitor cells (NG2 glia), which represent one of the most proliferative cell types in the brain, were affected by the PF4 treatment. Quantification of NG2+ cells revealed no differences in either the subgranular zone or the granular cell layer of the dentate gyrus (Supplementary Fig. 9a, b), suggesting that NG2 glia in this region are not responsive to changes in systemic PF4 levels. Phenotyping of proliferating Ki67+ cells with the early and intermediate neural progenitor cell markers Sox2 and Tbr2 revealed that most of these cells (combined ~90% in both treatment groups) were Sox2+Tbr2− or Sox2+Tbr2+, confirming that they belonged to the neuronal lineage (Fig. 5d). Very few proliferating cells were Sox2−Tbr2+ (1.3% ± 1.9% in saline-treated mice vs. 3.0 ± 2.4% in PF4-treated animals). We also observed no significant differences in the relative proportion of the cells expressing each marker between PF4-treated and control mice (Fig. 5d). However, when we compared the proportion of proliferating neural precursor cells that were at a later developmental stage, we observed an increase in Ki67+ DCX+ immature neurons following PF4 treatment (Fig. 5e), suggesting an effect on this more mature population. This is in accordance with the increase in immature neurons that we observed in the dentate gyrus of aged sedentary mice following PF4 supplementation (Fig. 5f, g). In contrast, we found no difference in the number of proliferating cells in the subventricular zone, suggesting that this effect is specific to adult hippocampal neurogenesis (Supplementary Fig. 9c, d).

As ageing is known to reduce total dendritic length and to decrease dendritic complexity[27], we also investigated whether PF4 could rejuvenate dendritic architecture (Fig. 5h). A branch tracing analysis revealed that the DCX+ cells in PF4-treated mice had elongated dendritic trees, with significant increases in total dendritic length, length of the primary dendrite and length of the longest dendrites (Fig. 5i, j), whereas the number of branches and branch points were not affected (Fig. 5k). We also determined whether the lack of PF4 influences the dendritic arborisation of newborn neurons by performing dendritic branch tracing on DCX+ cells in PF4 knockout mice (Fig. 5l). Although we observed no differences in overall length and complexity, the dendritic trees of the DCX+ cells in mice lacking PF4 had significantly shorter primary dendrites than those of their wildtype littermates (Fig. 5m, n). This identifies a role of PF4 in dendritic morphogenesis and corroborates our finding that PF4 promotes the maturation of newborn neurons. We also investigated the dendritic architecture of DCX+ cells following 28 days of exercise. Similar to the immature neurons of PF4-treated mice, those of exercising animals exhibited a significant increase in the total dendritic length (Supplementary Fig. 10a, b). However, although PF4 promoted the elongation of dendrites, the increase in dendritic length in aged exercising mice could be attributed to increases in branch complexity (Supplementary Fig. 10c).

### The exerkine PF4 rejuvenates cognitive function in aged mice

To determine whether systemic PF4 administration could recapitulate the rejuvenating effects of exercise on hippocampus-associated learning and memory, we systemically administered PF4 or saline for 24 days to 20-month-old mice as described above, and subsequently tested the mice in the novel object location and contextual fear conditioning tasks (Fig. 6a, b, e). During training in the novel object location task, the mice were placed in a chamber containing two identical objects, which they were free to explore for 10 min 24 h later, the long-term memory of the animals was assessed in a test session, during which one of the objects was moved to a novel location. Whereas mice from both groups spent a similar amount of time exploring both objects during the test session (Fig. 6c), PF4-treated animals displayed an improved recognition of the novel location, as evidenced by an increase in the time they spent at the novel location as well as more frequent visits to the moved object (Fig. 6d), thereby suggesting an improvement in long-term memory capacity. PF4 treatment did not induce differences in the overall movement, velocity, or weight of the mice (Supplementary Fig. 11a–c).

We next assessed the contextual memory of the mice using contextual fear conditioning. In this task, the mice were placed in a novel environment for 10 min, during which four 1 s, 0.8 mA foot shocks were delivered at 2, 4, 6 and 8 min. Although both groups displayed acquisition of conditioned fear during the training session, PF4-treated animals exhibited increased freezing behaviour compared to saline-treated mice, indicating improvements in working memory capacity (Fig. 6f). On the testing day, 24 h later, the mice were placed into the same fear conditioning chamber containing the same context, but without the delivery of foot shocks. Analysis of their freezing behaviour revealed that the PF4 supplementation also improved contextual fear memory, as PF4-injected mice displayed a higher freezing score compared to control mice (Fig. 6g).

To confirm the PF4-mediated improvements in hippocampus-related learning and memory, we repeated the injection paradigm in a separate cohort of mice and subsequently tested the animals in the active place avoidance task (Fig. 6h, i). In this test, the mice are placed on a rotating platform and, using spatial cues, must learn to avoid a stationary shock zone where they receive a mild foot shock[28]. Whereas young mice perform well in this test, aged animals fail to learn the task[21]. However, systemic PF4 treatment of aged mice ameliorated this cognitive deficit, as evidenced by a significant reduction in the number of entrances into the shock zone (Fig. 6j, k), as well as significant reduction in the number of shocks received and entries into the shock zone relative to the distance travelled (Fig. 6l, m). The PF4-treated mice also showed a continuous increase in the time before their first and second entrances into the shock zone, whereas the performance of saline-treated animals plateaued after 2–3 days of testing (Fig. 6n, o). The PF4 treatment did not impact the overall distance covered during the task, or the speed or weight of the mice (Supplementary Fig. 11d). Together these results reveal that systemic PF4 treatment can recapitulate the beneficial effects of exercise by rejuvenating hippocampal neurogenesis and restoring cognitive function in the aged brain.

### The rejuvenating effects of PF4 are neurogenesis-dependent

Finally, to gain a mechanistic insight into the cognition-enhancing effects of PF4, we used a DCX^DTR transgenic mouse model[28], which we aged to 20-months old. These mice express the human diphtheria toxin receptor (DTR) on the surface of DCX+ cells, leading to the selective ablation of newborn neurons upon administration of diphtheria toxin (DT), whereas other cell types remain unaffected[28]. Following the same experimental timeline as for C57BL/6J wildtype

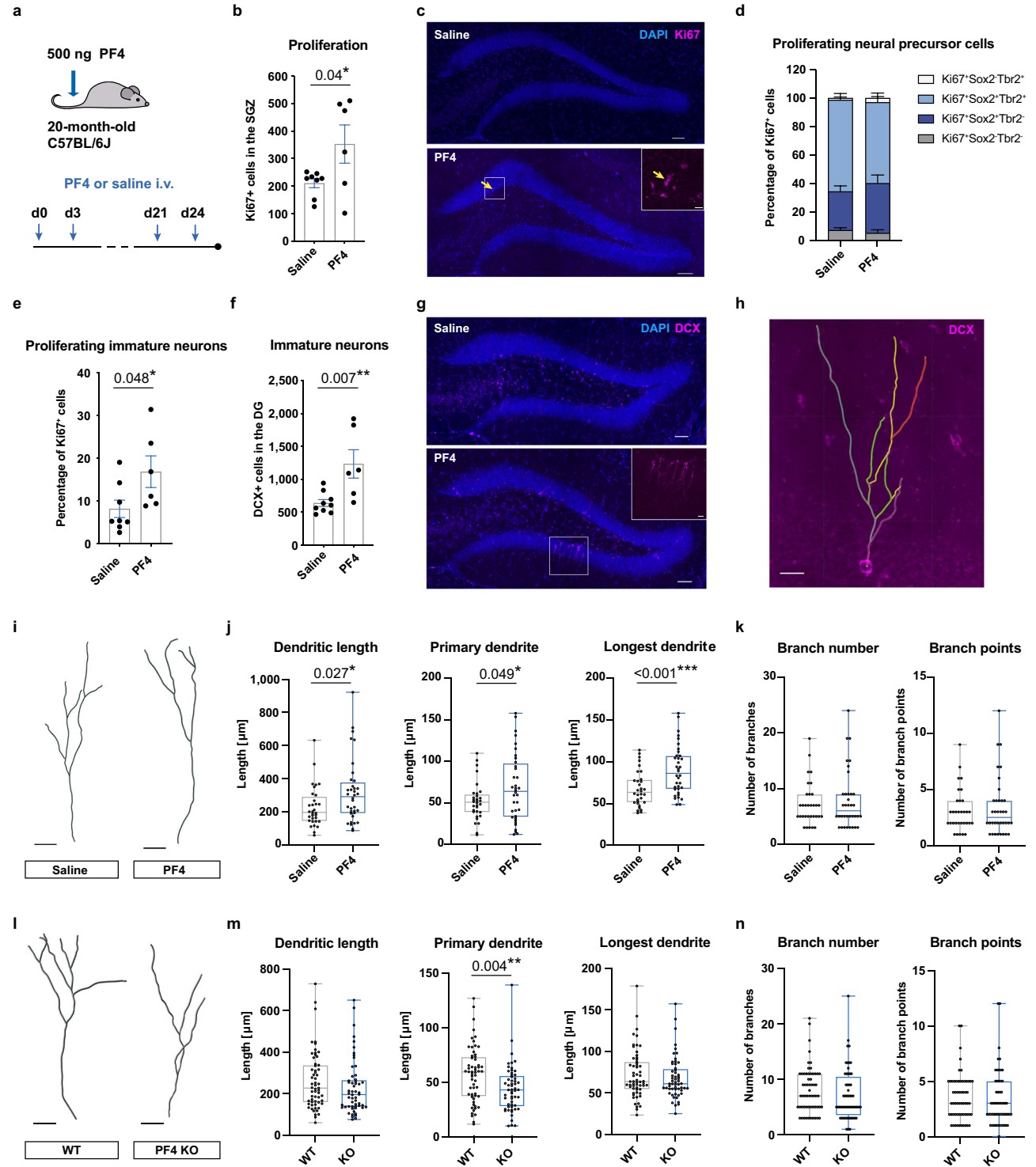

**Fig. 5 | Systemic PF4 increases adult hippocampal neurogenesis in aged mice.**
**a** Experimental design. **b** 20-month-old C57BL/6J mice receiving intravenous (i.v.)
PF4 injections showed significant increases in Ki67+ cells in the subgranular zone
(SGZ; saline *n* = 8 mice; PF4 *n* = 6 mice). **c** Representative image of Ki67+ cells. The
yellow arrow shows a trio of proliferating cells. Scale bars: 50 μm and 10 μm in box
highlighting three proliferating cells. **d** Phenotyping of the Ki67+ cells in saline- and
PF4-treated mice (saline *n* = 8 mice; PF4 *n* = 6 mice). PF4 increased the percentage
of proliferating DCX+ cells (**e**) and the total number of DCX+ cells (**f**) in the dentate
gyrus (DG) compared to saline-treated controls (saline *n* = 8 mice (**e**) and *n* = 9 mice
(**f**); PF4 *n* = 6 mice). **g** Representative images of DCX staining. Scale bar: 50 and
15 μm in box highlighting a cluster of DCX+ cells. **h** Representative image of den-
dritic branch tracing, showing a postmitotic DCX+ cell. Scale bar: 15 μm.
**i** Representative reconstruction of DCX+ postmitotic cells. Scale bars: 20 μm.

**j** Postmitotic DCX+ cells of mice receiving PF4 showed increased total dendritic
length, longer primary dendrites, and length of the longest dendrite per traced cell
(saline *n* = 32 cells; PF4 *n* = 36 cells). **k** PF4 did not affect dendritic branch numbers
or the number of branch points (saline *n* = 32 cells; PF4 *n* = 36 cells).
**l** Representative reconstruction of DCX+ postmitotic cells. Scale bars: 15 μm. **m** PF4
KO mice displayed significantly shorter primary dendrites in the postmitotic DCX+
cells compared to WT littermates (WT *n* = 59 cells; PF4 KO *n* = 56 cells). **n** Dendritic
complexity was unaffected in mice lacking PF4 (WT *n* = 59 cells; PF4 KO *n* = 56 cells).
Bars represent mean ± SEM. Box plots show median ± interquartile range, with
whiskers defining minimum and maximum. Statistical analysis was performed using
unpaired Student's two-tailed *t* tests. *$p < 0.05$, **$p < 0.01$, ***$p < 0.001$. Source data
are provided as Source Data file.

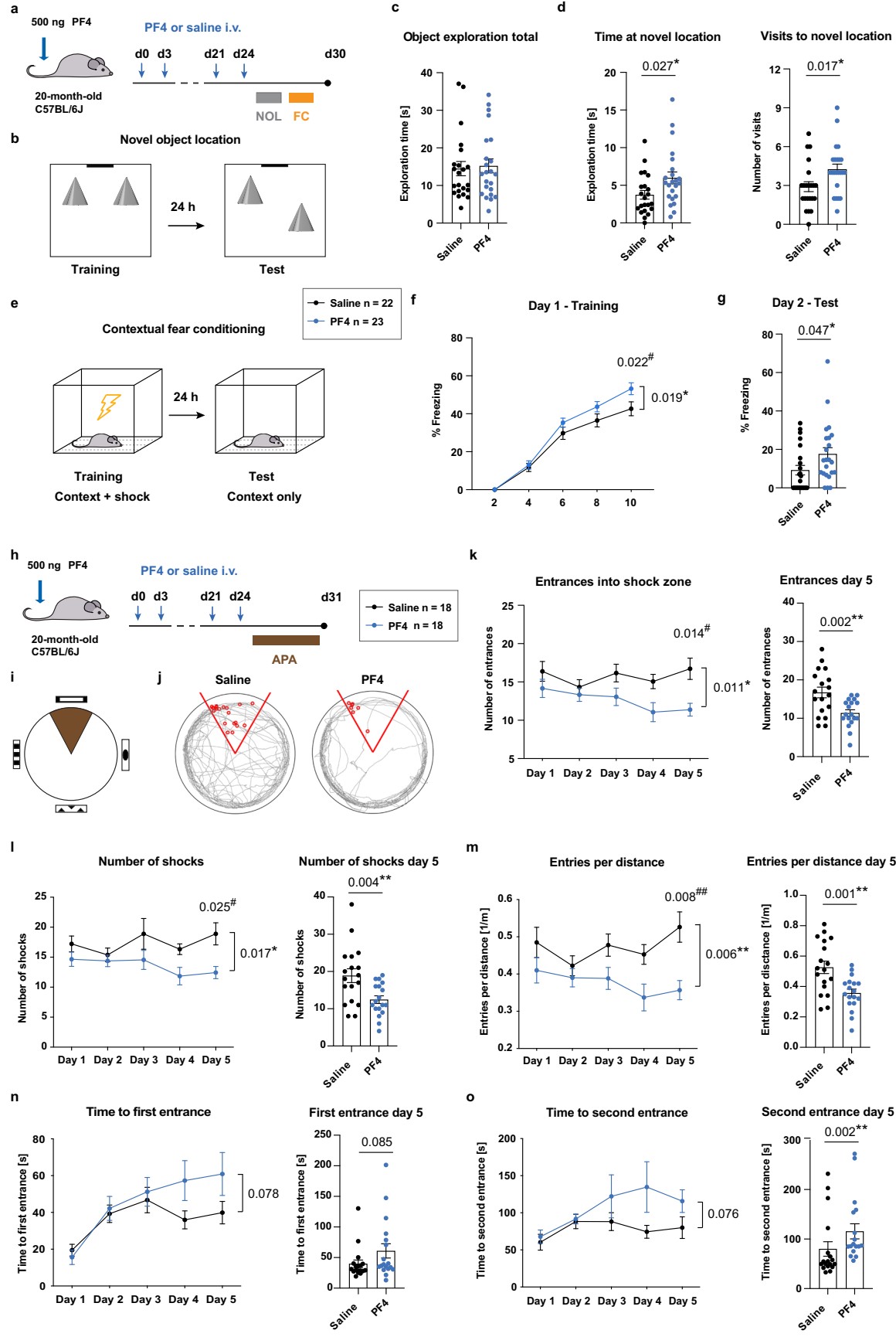

**Fig. 6 | Systemic administration of PF4 rescues hippocampal learning and memory in aged mice. a** Experimental design of the novel object location (NOL) and fear conditioning (FC) tasks. **b** Schematic illustration of the NOL arena. **c** Mice from both treatment groups explored both objects during the first minute of the test session (saline *n* = 22 mice; PF4 *n* = 23 mice). **d** PF4-treated mice (blue) spent more time at the novel location and visited the moved object more frequently (saline *n* = 22 mice; PF4 *n* = 23 mice). **e** Schematic illustration of the FC paradigm. **f** PF4-treated mice showed a higher freezing response in the FC task (saline *n* = 22 mice; PF4 *n* = 23 mice). **g** PF4-treated mice showed higher freezing behaviour in the fear recall test (saline *n* = 22 mice; PF4 *n* = 23 mice). **h** Experimental design of the active place avoidance (APA) task. **i** Schematic illustration of the APA arena with angled shock zone (grey) and visual cues. **j** Representative paths of mice in the APA

arena, with red circles indicating the shocks received when entering the shock zone (red lines). **k** PF4-treated mice (blue) showed a reduction of entrances into the shock zone compared to saline-treated mice (black). Performance of individual mice on day 5 is shown on the right (*n* = 18 mice per group). Further improvements were observed in the number of shocks (**l**), entries into the shock zone per distance travelled (**m**), and time to first (**n**) and second (**o**) entrance into the shock zone (*n* = 18 mice per group). Bars represent mean ± SEM. Statistical analysis was performed using unpaired Student's two-tailed *t* tests in (**d**, **g**, **k**–**m**), and two-tailed Mann–Whitney *U* tests in (**n**, **o**) (right panels, \**p* < 0.05, \*\**p* < 0.01). Two-way repeated measures ANOVAs with Sidak post hoc comparison were performed in (**f**, **k**–**o**) (left panels, effect of treatment \**p* < 0.05, \*\**p* < 0.01, Sidak comparison #*p* < 0.05, ##*p* < 0.01). i.v. intravenous. Source data are provided as Source Data file.

animals, aged DCX$^{DTR}$ mice received PF4 injections every third day over a period of 24 days and were subsequently tested in the active place avoidance task. On days 18, 20, 22 and 24, a subset of the mice received a single injection of DT, resulting in a significant reduction in the number of DCX$^+$ cells compared to PF4-treated mice which received vehicle (Fig. 7a, b). Again, we saw no changes in the overall distance covered during the task, the speed or weight of the DCX$^{DTR}$ mice following systemic PF4 treatment (Supplementary Fig. 11e). Similar to aged C57BL/6J mice, the PF4-treated nonablated DCX$^{DTR}$ mice exhibited a significant improvement in the active place avoidance task in comparison to saline-treated control mice (Fig. 7c–i), including continuous increase in the time spent avoiding the shock zone before entering it for the first or second time (Fig. 7g, h). In contrast, the cognition-enhancing effects of PF4 were absent after the ablation of DCX$^+$ cells with DT, highlighting the necessity of neurogenesis for the PF4-mediated cognitive enhancement (Fig. 7c–i).

## Discussion

Together, our data demonstrate that the cognition-enhancing benefits of exercise on the aged brain are mediated by platelets and can be replicated through the systemic administration of the platelet-released exerkine PF4 in an adult hippocampal neurogenesis-dependent manner.

Although the presence of neurogenesis in the hippocampus of adult humans has been hotly debated, the current overwhelming consensus in the field is that this process continues throughout life but declines with age, as well as in individuals with Alzheimer's disease[29–32]. Given that ageing is the most dominant risk factor for dementia-related neurodegenerative diseases, it is imperative to develop novel strategies to maintain cognitive integrity in the elderly. One possibility is to harness the regenerative capacity of stem cells. Physical exercise is one of the strongest enhancers of adult hippocampal neurogenesis[4,5], and older adults who perform regular physical activity are more likely to maintain their cognitive ability[1,3]. However, we still have only a limited understanding of how the systemic effects of exercise are communicated to the brain. Exerkines, which are released from multiple cells or organs, are emerging as likely mediators of this response. Examples of exerkines that are involved in mediating brain plasticity include the liver-derived protein glycosylphosphatidylinositol (GPI)-specific phospholipase D1 (Gpld1)[6], the myokine cathepsin B[8] and proteins in the complement and coagulation pathways, including clusterin, which the authors suggested is released from hepatocytes or cardiomyocytes[7]. Although our proteomic analysis revealed that these three exerkines are also present in the platelets of both young and aged mice, their levels remained unaffected by exercise, further suggesting that multiple sources of exerkines contribute to exercise-mediated brain rejuvenation. Brain-derived neurotrophic factor (BDNF) is also known to be involved in the exercise-mediated increase in adult hippocampal neurogenesis[33]. However, mouse platelets do not carry BDNF[34] and it is therefore unlikely that this factor regulates the platelet-mediated neurogenic response to exercise that we observed in mice. This is in line with proteomics analyses performed by ourselves

and others[35], which found no detectable BNDF protein in the platelets of young or aged mice.

Our results provide the first evidence that platelets release exerkines, including PF4, that increase adult neurogenesis and rejuvenate cognitive function, thereby cementing their important role in the regulation of brain function. Platelets are attracting attention as a biotherapy due to their amazing reservoir of potent anti-inflammatory, neurotrophic and antioxidant molecules. Platelets, in the form of platelet-rich plasma or platelet lysates, are widely used in many regenerative applications, including post-surgical healing[36], treatment of musculoskeletal injuries and osteoarthritis[37–39], and skin rejuvenation[40]. The synergistic neuroprotective effect of platelet-derived factors has also recently been evidenced in mouse models of traumatic brain injury[41], amyotrophic lateral sclerosis[42] and Parkinson's disease[43].

In the present study we identify PF4 as an exerkine that is both sufficient and necessary for the exercise-mediated increase in adult hippocampal precursor cell proliferation. PF4 is a small chemokine whose roles in the periphery include coagulation[44], inflammation[45–47] and wound repair[46,48]. However, less is known about its function in the brain, and the cellular mechanisms through which PF4 exerts its beneficial effects on neural precursor cells remain unclear. Our data exclude the possibility that PF4 recruits quiescent neural precursor cells into proliferation, suggesting instead that it promotes the neurogenic process at later stages, such as during neuronal differentiation, integration or survival. Although we also observed PF4-induced increases in neural precursor cell proliferation in aged animals, this increase was primarily attributed to proliferating DCX$^+$ neural precursor cells, suggesting an effect of PF4 on this more mature neural precursor cell population. This is supported by our finding that PF4 rejuvenates the dendrite morphology of newborn neurons in the aged brain by stimulating dendritic elongation, with neurite outgrowth proposed to provide advantages for the survival of maturing neurons as well as their functional integration into the local circuitry[49]. In contrast to aged mice, we observed no increase in proliferation in the dentate gyrus of young animals. One possible explanation for the difference observed between the young and old mice is that aged mice exhibit extremely low levels of baseline proliferation. Therefore, even very small increases in the absolute number of proliferating cells following PF4 treatment have a significant effect on the total cell count. Such effects could, however, be masked in 8-week-old animals, in which the levels of baseline proliferation are ~40 times higher than in 20-month-old mice.

A recent study reported a direct communication route between the blood and neural precursor cells in the hippocampus via the vessel-associated apical processes of radial glia-like stem cells, thereby allowing the exchange of systemic factors with these cells[50]. Although the detailed mechanisms through which PF4 affects the hippocampal stem cell niche remain to be determined, we propose that the exerkine confers the cognition-enhancing benefits of exercise by directly inducing pro-neurogenic changes. Notably, our data reveal that PF4 specifically stimulates adult neurogenesis in the dentate gyrus,

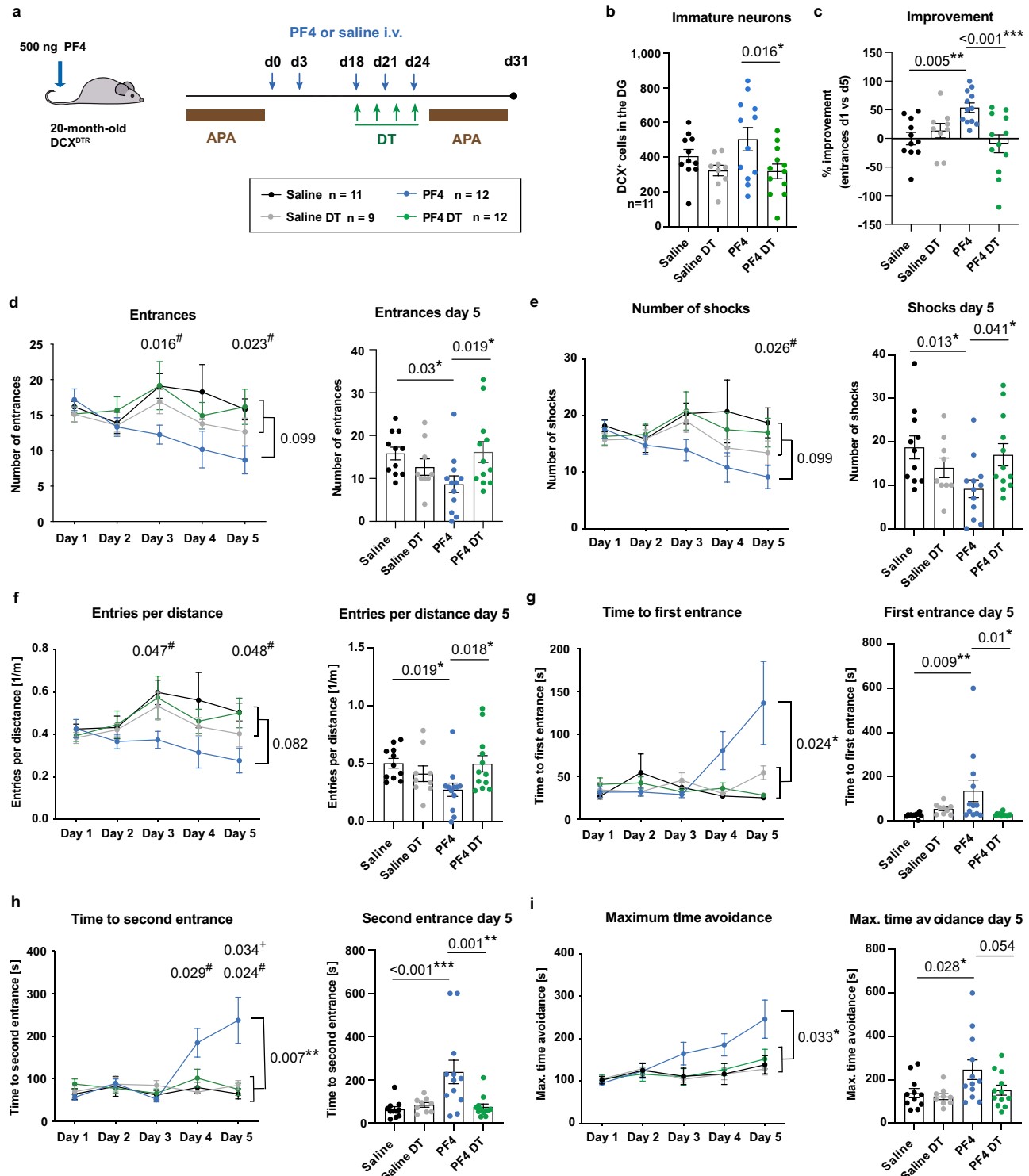

**Fig. 7 | The beneficial effect of PF4 on cognition is neurogenesis-dependent.**
**a** Experimental design of aged DCX[DTR] mice receiving intravenous (i.v.) PF4 injections every third day for 24 days and subsequent testing in the active place avoidance (APA) task, with an additional 4 injections of diphtheria toxin (DT) beginning on day 18. **b** The number of DCX[+] cells following DT treatment and APA testing. **c** PF4-treated mice (blue) improved during the APA task, whereas this effect was absent in all other groups, including when PF4-treated mice received injections of DT, specifically ablating adult neurogenesis. **d** PF4-treated mice showed a reduction in the number of entries into the shock zone, whereas mice in the other three groups did not improve. Improvements following PF4 treatment were observed in all parameters of the test, including the number of shocks received (**e**), entries into the shock zone per distance travelled (**f**), and time to first (**g**) and second (**h**) entrance into the shock zone, as well as the maximum time spent avoiding the shock zone (**i**). Saline $n = 11$ mice; Saline DT $n = 9$ mice; PF4 $n = 12$ mice; PF4 DT $n = 12$ mice. Bars represent mean ± SEM. Statistical analysis was performed using one-way ANOVAs with Sidak post hoc comparison to compare groups on day 5 and two-way repeated measures ANOVAs with Dunnett's post hoc comparison to compare groups throughout the 5-day test. ANOVA result *$p < 0.05$, **$p < 0.01$, ***$p < 0.001$, Dunnett's comparison Saline vs. PF4 #$p < 0.05$, PF4 vs. PF4 DT +$p < 0.05$. Source data are provided as Source Data file.

consistent with the effect of exercise which promotes neurogenesis in the hippocampus but not the subventricular zone[15]. However, we do not exclude the possibility of additional mechanisms by which PF4 may enhance cognition, either by acting directly on other brain cell types, or through indirect peripheral effects.

Platelets are proficient at sensing, adapting, and responding to environmental stimuli. Our data demonstrate that platelets from both young and old mice respond to exercise. Interestingly, however, the proteomic signatures highlight that this response differs from that of other classical platelet activation pathways, such as wound healing or the response to infection. Although our proteomic analysis revealed distinct protein signatures at different running timepoints and ages, we did observe conserved changes, including proteins primarily involved in immune responses and redox regulation. In a previous study we reported that the exercise-triggered recruitment of neural precursor cells into proliferation is characterised by a shift from a high reactive oxygen species (ROS) content to moderate cellular ROS levels[10]. In support of this, we further demonstrated that an increase in the systemic transport of the antioxidant selenium underlies the exercise-induced activation of neural precursor cells, concomitant with a reduction in cellular ROS levels in these cells[51]. However, the source of the increased selenium transport proteins in the plasma remains unknown. In the present study, we identified a range of well-known antioxidants, including two selenoproteins and sulfiredoxin, among the proteins which were increased in the platelets of aged mice following short periods of exercise (Supplementary Data 7). As this timepoint precedes that at which we observed a neurogenic response to exercise in aged animals, our results suggest that platelet-mediated increases in systemic selenium transport could underlie the exercise-induced activation of neural precursor cells.

Accumulating evidence demonstrates a clear link between an active lifestyle and brain health, with exercise known to ameliorate cognitive decline in ageing and neurodegenerative conditions[1–3]. Moreover, several human studies report benefits of exercise on the cardiovascular system[52,53]. Although our study describes the exercise-induced platelet activation response as beneficial, it is worth noting that several neurodegenerative conditions, as well as cardiovascular events, are linked to chronic platelet activation responses[54,55]. Interestingly, regular endurance training in humans has been shown to reduce basal levels of platelet reactivity, including their adhesiveness and aggregability, suggesting that chronic exercise has protective effects against cardiovascular disease[56]. In this study we have characterised a platelet activation response to exercise in young and aged mice which appears to be different from other classical platelet activation events, and have identified PF4, which is released from exercise-induced activated platelets, as an anti-geronic exerkine that rejuvenates neurogenesis and cognition in the aged brain. These findings highlight the potential value of novel therapeutics based on systemically delivered exerkines.

## Methods
### Animals
Young adult female C57BL/6J mice (8–10 weeks old) were obtained from the Animal Resources Centre Perth. Aged animals (C57BL6/J and DCX[DTR] mice[28]) between 18 and 20 months of age at the beginning of each experiment were obtained from aged mouse colonies maintained at the animal facility of the Queensland Brain Institute at The University of Queensland. Homozygous PF4 knockout mice were obtained from Anna Kowalska and have been described previously[14]. If not otherwise stated, the mice were group-housed on a 12 h light/dark cycle and had ad libitum access to food and water, with their housing temperature ranging from 18.5 to 24 °C and 40 to 60% humidity. Following perfusion of aged mice, a careful examination of organs was performed, leading to the exclusion of animals with indications of chronic inflammation, such as a greatly enlarged spleen, lymph nodes and liver, as well as tumour growth. All experiments were approved by The

University of Queensland Animal Ethics Committee (2018/AE000549, 2021/AE000317, 2023/AE000013) and performed in accordance with the Australian Code of Practice for the Care and Use of Animals for Scientific Purposes or in accordance with institutional guidelines approved by the University of California San Francisco Institutional Animal Care and Use Committee (AN194088).

### Running paradigms
Experiments with C57BL/6J mice were performed at The University of Queensland, with mice housed in cages with ad libitum access to a hanging running wheel (Labodia; 11 cm in diameter). Young mice (short-term running; 8 weeks old) were single housed for the running duration in order to acquire accurate wheel usage information, whereas aged mice (longer periods; 18 months old) were housed three per cage to avoid adverse effects on the behaviour of the mice which could be caused by long-term social isolation. Age-matched control mice were housed under the same conditions in cages without a running wheel. To address neuronal survival in aged mice following 28 days of running, the mice received three intraperitoneal injections of 5-bromo-2′-deoxy-yuridine (BrdU; 50 mg/kg) 24 h apart, before being allocated to running or standard cages. Experiments with PF4 knockout mice (male, 8 weeks old) were performed at the University of California San Francisco, with mice housed individually for 10 days in cages with ad libitum access to a low-profile mouse running wheel (Med Associates; cat # ENV-047).

### Neural precursor cell culture
A neural precursor cell line was generated from the dentate gyri of 8-week-old male C57BL/6JRj mice and maintained as an adherent monolayer culture as described previously[13]. For the experiments, neural precursor cells from an 80% confluent culture were seeded into poly-D-lysine (PDL)/laminin-coated wells with or without coverslips at a density of $2 \times 10^4$ cells/cm² and cultured for 48 h in growth medium (neural basal medium containing 2% B-27® supplement (50X), 1% penicillin/streptomycin (10,000 U/ml) and 1% GlutaMAX™ (100X)) containing 20 ng/ml epidermal growth factor (EGF) and 20 ng/ml fibroblast growth factor 2 (FGF-2). Recombinant mouse PF4 (Prospec, cat# chm-245, lot# 1113PMPF4) was reconstituted in sterile 0.9% sodium chloride and aliquots of 100 μg/ml stock frozen for each experiment. Each independent experiment was performed in triplicate on a separate consecutive passage of the cells.

### Proliferation assay
The proliferation assay was performed according to previously published protocols[13]. In brief, the proliferation medium was supplemented with 100 or 500 ng/ml mouse recombinant PF4 protein (Prospec cat# chm-245) or an equal volume of 0.9% sodium chloride. After 48 h, BrdU was added to each well (final concentration 10 μM) and the cells were incubated for 2 h at 37 °C. Following this, the medium was removed, and the wells were washed once with 0.01 M phosphate buffered saline (PBS). They were then fixed with 4% paraformaldehyde solution for 12 min at room temperature and washed twice with 0.01 M PBS. A total of six independent experiments were performed.

### Differentiation assay
The differentiation assay was performed according to previously published protocols[13]. In brief, 48 h after seeding, the proliferation medium was replaced by growth medium containing 5 ng/ml FGF-2 and no EGF. In addition, mouse recombinant PF4 protein was added at 100 or 500 ng/ml. Controls were supplemented with an equal volume of 0.9% sodium chloride. After 48 h the medium was replaced by growth medium without growth factors, allowing differentiation of the cells. On day 9 of the assay, the differentiated cells were fixed with 4% paraformaldehyde in 0.1 M phosphate buffer and washed twice with 0.01 M PBS. A total of six independent experiments were performed.

## Immunostaining of neural precursor cells

Immunostaining was performed on two randomly selected coverslips per condition. For BrdU labelling the coverslips were washed twice with 0.9% NaCl, followed by incubation with 1 M hydrochloric acid (HCl, Merck Millipore) at 37 °C for 30 min. The coverslips were then washed once with 0.1 M borate buffer and three times with 0.01 M PBS. For immunostaining of differentiated cells the above steps were omitted and the coverslips were washed once with 0.01 M PBS. Blocking was then performed for 1 h in 0.01 M PBS containing 0.01% sodium azide, 10% normal goat serum and 0.1% Triton X-100. Incubation with primary antibodies rat anti-BrdU (1:500, AbD Serotec, cat# OBT0030, RRID AB_609568) or a combination of rabbit anti-GFAP (1:500, Dako Agilent, cat# Z0334, RRID AB_10013382) and mouse anti-β-III-tubulin (1:2000; clone 5G8, Promega, cat#G7121, RRID AB_430874) in 0.01 M PBS containing 0.01% sodium azide, 3% normal goat serum and 0.1% Triton X-100 was performed overnight at 4 °C. The coverslips were washed three times with 0.01 M PBS, followed by incubation with corresponding secondary antibodies (all 1:1000; goat anti-rabbit Alexa Fluor 488 (Thermo Fisher Scientific, cat# A11008, RRID AB_143165) and goat anti-mouse Alexa Fluor 568 (Invitrogen, cat# A11031, RRID AB_144696)) in 0.01 M PBS containing 0.01% sodium azide, 3% normal goat serum and 0.1% Triton X-100 at room temperature in the dark for 1 h. The coverslips were then washed with 0.01 M PBS, incubated with 4′,6-diamidino-2-phenylindole (DAPI; 1:5000 in PBS, Thermo Fisher Scientific) for 10 min, washed again with 0.01 M PBS, and finally dip-washed in dH$_2$O and mounted onto glass slides using fluorescence mounting medium (Dako Agilent).

## Quantification of proliferating and differentiated cells

Quantification of proliferating and differentiated cells was performed on 5 random fields of view (FOVs) on each of the two stained coverslips, making a total of 10 FOVs per condition. FOVs were determined in the DAPI channel. The images were acquired at ×200 magnification, capturing an average of 120 cells per FOV, using a Zeiss AxioImager Z1 microscope and Zeiss ZEN software (blue edition). BrdU$^+$, β-III-tubulin$^+$, GFAP$^+$ and DAPI$^+$ cells were counted using Adobe Photoshop®. The proportion of proliferating or differentiated cells per experiment was determined by quantifying the number of BrdU$^+$ cells or β-III-tubulin$^+$ and GFAP$^+$ cells relative to the total number of cells (DAPI$^+$ cells), with one data point referring to the average value of 10 FOVs. A total of ~1200 cells per condition were analysed.

## Click-iT® EdU Flow Cytometry Proliferation Assay

Neural precursor cells were cultured under proliferation conditions in PDL/laminin-coated 6-well plates with recombinant mouse PF4 protein diluted in sterile 0.9% sodium chloride. Control cells were cultured with an equal volume of sterile 0.9% sodium chloride. After 48 h, the cells were incubated with EdU (final concentration 10 μM) for 2 h and the Click-iT® EdU Flow Cytometry Cell Proliferation Assay (Thermo Fisher Scientific cat# C10634) was performed according to the manufacturer's instructions. The cells were also incubated with Hoechst 33342 (1:5000 in saponin-based permeabilization and wash reagent) for 10 min in the dark. They were then transferred into a flow cytometry tube and analysed using an LSRII flow cytometer and FACSDiva software (v9.0; BD Biosciences) on the same day. For this, viable cells were first determined using forward scatter and side scatter. Next, doublets were excluded from single cells by plotting Hoechst-width against Hoechst-area. Finally, to define the cell cycle phases, the DNA content (Hoechst-area) was plotted against the EdU signal. A total of 50,000 single cell events were recorded by flow cytometry. Data analysis was performed using FlowJo software (BD, v10.8.1). A total of seven independent experiments were performed with three technical replicates each.

## PF4 labelling and uptake study

Recombinant mouse PF4 protein was fluorescently labelled via Lightning-Link® technology using the Alexa Fluor® 568 Conjugation Kit (cat# ab269821). Accounting for the size and amount of protein, modifier reagent was first added to 10 μl of 100 μg/ml PF4, which was then mixed with 10 μl of lyophilised Alexa Fluor® 568 and incubated for 15 min in the dark at room temperature. Following this, quencher was added to the mix to wash away any remaining free label. The labelled protein was added to adherent monolayer cells which had reached 60% confluency and cultured for either 2, 6 or 24 h, after which the cells were fixed with 4% paraformaldehyde solution and washed twice with 0.01 M PBS. A total of six independent experiments were performed. Cells were stained with DAPI and imaged using a Yokogawa W1 spinning disk confocal microscope at ×1000 magnification and SlideBook software v6.0.

## Platelet count and platelet activation state

Eighteen-month-old female C57BL/6J mice were housed in groups of three for 35 days in cages with or without a hanging running wheel (Labodia; 11 cm in diameter). Platelet count and activation state were measured on days 1, 4, 8, 15, 21, 28 and 35 using flow cytometry. For this, 5 μl of blood were collected from the tail tip of the same mice on each testing day and transferred into 45 μl of 0.11 M sodium citrate (Merck) in 0.01 M PBS, pH 6.5, and mixed carefully. Ten μl of the diluted whole blood were stained with Armenian hamster anti-CD61-PE (1:25; clone 2C9.G3; Thermo Fisher Scientific, cat# 12-0611; RRID AB_465718) and mouse anti-CD62P-APC (1:4; clone RMP-1; Biolegend, cat# 148304; RRID AB_2565273) for 30 min at room temperature in the dark. Prior to the experiment, the antibodies were titrated to determine the optimal dilutions for each lot. To determine the platelet count, 10 μl of Precision Count Beads (Biolegend, cat# 424902) were added to the samples, before fixation with platelet fixation solution (final volume 1 ml; 0.01 M PBS containing 0.2% bovine serum albumin (BSA; Sigma-Aldrich), 0.1% D-glucose (Sigma) and 0.1% formalin (Labserv)). A total of 100,000 platelet events were recorded by flow cytometry using a BD LSRII flow cytometer. Platelets were defined based on forward scatter and side scatter, followed by doublet depletion and gating on forward scatter and the expression of CD61. Unstained platelet preparations as well as single stained samples were included to determine the level of autofluorescence and verify the placement of the gates. The platelet count was determined using a defined number of counting beads, which were counted together with the CD61$^+$ cells. The flow cytometry-measured values of the platelet count were confirmed using an automated haematology analyser (Mythic™ 18 Vet) on 5 mice per group throughout the first three time points of the experiment. Activated platelets were determined as the percentage of CD62P$^+$ cells within the CD61$^+$ population. The platelet count and activation state were measured from ten individual animals per time point. Each measure of platelet activation was normalised to the mean value obtained from the standard-housed mice on each day. Data analyses was performed using the FlowJo software (BD, v10.8.1).

## Corticosterone measurements in whole blood

Five μl of blood were collected from the tail tip of 18-month-old exercising or standard-housed female C57BL/6J mice and transferred into 45 μl of 0.11 M sodium citrate (Merck) in 0.01 M PBS, pH 6.5. Blood and buffer were mixed carefully, and the samples stored at −80 °C until corticosterone levels were measured using the corticosterone ELISA kit from Enzo (ADI-900-097). A sample titration was performed prior to the measurements to determine an optimal sample dilution of 1:800.

## Tail vein injections

PF4 (Prospec; cat# chm-245; lot# 1113PMPF4 and lot# 717 PMPF4) was resuspended with 0.9% sterile sodium chloride to a stock

concentration of 50 µg/ml and stored as aliquots at −20 °C. Immediately prior to the injections, the PF4 stock was thawed and diluted to 5 µg/ml with 0.9% sterile sodium chloride. In total, 100 µl of the diluted PF4 were injected into the lateral tail vein of 8-week-old and 20-month-old female C57BL/6J mice or 20-month-old male and female DCX$^{DTR}$ mice, alternating between the left and the right lateral tail veins. The control group received 100 µl of 0.9% sodium chloride. Prior to the injection, the cages were placed under a ceramic light heat lamp for ~3 min. The mice were placed in an animal restrainer and the tail was swabbed with 70% ethanol. Tail vein injections were performed using a 27-gauge needle. Young mice were injected every second day over the course of 1 week, making 4 injections in total. Aged mice received injections every third day for 24 days, making a total of nine injections.

### Double labelling paradigm with CldU and IdU

To label all proliferating neural precursor cells in 8-week-old female C57BL/6J mice, three intraperitoneal injections of 5-chloro-2′-deoxyuridine (CldU; 42.5 µg/kg) were given 9 h apart. One hour after the last CldU injection, the mice received a single intravenous injection of 100 µl PF4 (5 µg/ml). To distinguish cells that were newly recruited into proliferation from cells that were already proliferating prior to PF4 administration, three doses of 5-iodo-2′-deoxyuridine (IdU; 57.5 mg/kg) were given intraperitoneally, 9 h apart. The mice were perfused 1 h after the last IdU injection. Brains were prepared for histological analysis, and immunohistochemistry for CldU$^+$ and IdU$^+$ cells was performed as described in the immunohistochemistry section. Total numbers of CldU$^+$ and IdU$^+$ cells were quantified using a Zeiss Axiolmager Z1 microscope. Random fields of view were determined in the DAPI channel and at least 100 cells were phenotyped according to their CldU and IdU label. With this approach, cells that were already proliferating prior to the PF4 treatment and remained proliferative thereafter (CldU$^+$IdU$^+$), cells that started proliferating after PF4 treatment (CldU$^-$IdU$^+$), as well as cells that lost their proliferative phenotype during the course of the experiment (CldU$^+$ IdU$^-$) could be distinguished.

### Subcellular fractionation and western blotting

Eight-week-old female C57BL/6J mice were treated with either saline or PF4 (100 µl; 5 µg/ml), every second day for 1 week by i.p. injection (4 injections in total). Dentate gyri were collected and lysed in ice-cold sucrose buffer (0.32 M sucrose, 10 mM HEPES, pH 7.4). The homogenate was centrifuged at $1000 \times g$ for 10 min at 4 °C, yielding the supernatant fraction and the nuclear-enriched pellet. An aliquot of the homogenate was taken for further analysis as the total protein fraction, with the remainder being centrifuged at $10,000 \times g$ for 15 min at 4 °C to obtain a crude synaptosomal (P2) fraction. All fractions were denatured after adjusting total protein concentrations, followed by western blotting. Membranes were probed with specific antibodies against PSD-95 (1:2000; clone K28/43, NeuroMab, cat# 75-028, RRID AB_2292909), synaptophysin (1:5000; clone 7.2, Synaptic Systems, cat# 101011, RRID AB_887824) and β-actin (1:5000; clone 13E5, Sigma, cat# A5441, RRID AB_476744). Signals were developed with an enhanced chemiluminescence (ECL) method. Images were acquired on the Odyssey Fc imaging system (LI-COR) and band intensities were quantified using Image Studio Lite software (LI-COR).

### Neurite complexity of mature neurons

Primary hippocampal neurons were isolated from male and female embryonic day 17 C57BL/6J mouse brains. Primary hippocampal neurons were transfected at 13 days in vitro with a plasmid that encodes mEmerald fluorescence protein using Lipofectamine 2000 (Invitrogen) for 72 h. Neurons were treated with either sodium chloride or 100 ng/ml of mouse recombinant PF4 protein for 24 h. Fixed neurons were imaged with a ×63 oil-immersion objective on a Zeiss LSM510 confocal microscope. Series of optical sections were collected at

0.38 µm intervals, and maximal intensity projection was performed. Sholl analysis was performed using ImageJ image analysis software (v2.1.0/153c, National Institutes of Health) on thresholded images.

### Perfusion and brain tissue preparation for histology

Mice were deeply anaesthetised with 100 mg/kg ketamine (Ceva Animal Health)/20 mg/kg xylazine (Ilium) in 0.9% NaCl and perfused with 0.9% NaCl. The perfused brains were removed from the skull and postfixed in a 4% paraformaldehyde solution for 48 h at 4 °C. Before sectioning, the brains were stored in 30% sucrose (Sigma-Aldrich) for 48 h. In total, 40 µm coronal brain sections were cut on a sliding microtome (Leica SM2010 R) cooled with dry ice. Sections were collected and stored in cryoprotectant solution (CPS; 25% ethylenglycol (Carl Roth); 25% glycerol (VWR) in 0.1 M phosphate buffer) at 4 °C.

### Immunohistochemistry of free-floating brain sections

**Fluorescence staining for BrdU, CldU, IdU, Ki67, Sox2, Tbr2, DCX and NeuN.** Free-floating sections were transferred from CPS into 0.01 M PBS and washed twice with 0.01 M PBS containing 0.01% Tween-20, with which all washing steps were performed unless otherwise stated. For labelling of the thymidine analogues BrdU, CldU and IdU, the sections were washed twice with 0.9% NaCl and incubated for 30 min in 1.5 M HCl at 37 °C, followed by thorough washing. After one wash with dH$_2$O, antigen retrieval was performed using antigen recovery solution (2.6% sodium citrate, 3% sodium lauryl sulphate in dH$_2$O) for 10 min at 60 °C, followed by one wash each in dH$_2$O and 0.01 M PBS containing 0.01% Tween-20. Blocking was performed for 1 h in 0.01 M PBS containing 0.01% sodium azide, 10% normal goat serum and 0.2% Triton X-100. For antibody combinations which included goat anti-Sox2 antibodies, goat serum was replaced with horse serum in all blocking and antibody solutions. Incubation with primary antibodies (rat anti-BrdU (detects BrdU and CldU) clone BU1/75 (ICR1), 1:500, AbD Serotec, cat# OBT0030, RRID AB_609568; mouse anti-BrdU (IdU) clone B44, 1:500, BD, cat# 347580, RRID AB_10015219; rat anti-Ki67 clone SolA15, 1:1000, eBioscience, cat# 14-5698-82, RRID AB_10854564; rabbit anti-doublecortin (DCX) 1:2000, Cell Signaling Technology, cat# 4604, RRID AB_561007; rabbit anti-NeuN, 1:2,000, Abcam, cat# ab104225, RRID AB_10711153; rabbit anti-Tbr2 clone EPR19012, 1:800, Abcam, cat# ab183991, RRID: AB_2721040; goat anti-Sox2 1:200, R&D, cat# AF2018, RRID: AB_355110) diluted in antibody solution (0.01 M PBS containing 0.01% sodium azide, 3% normal goat serum and 0.2% Triton X-100) was performed for 48 h at 4 °C. Sections were thoroughly washed before incubation with corresponding secondary antibodies (all 1:1000; goat anti-rat Alexa Fluor 488, Jackson ImmunoResearch, cat# 112-545-167, RRID AB_2338362; goat anti-mouse Alexa Fluor 568, Invitrogen, cat# A11031, RRID AB_144696; goat anti-rabbit Alexa Fluor 647, Thermo Fisher Scientific, cat# A21245, RRID AB_2535813; goat anti-rabbit Alexa Fluor 568, Invitrogen, cat# A-11011, RRID AB_143157; donkey anti-goat Alexa Fluor 488, Thermo Fisher Scientific, cat# A11055, RRID AB_2534102; donkey anti-rabbit Alexa Fluor 647, Invitrogen, cat# A-31573, RRID AB_2536183; donkey anti-rat Alexa Fluor 594, Thermo Fisher Scientific, cat# A21209, RRID AB_2535795) for 4 h. They were then washed twice with 0.01 M PBS and incubated with DAPI (1:5000 in 0.01 M PBS) for 10 min. After two additional washes, the sections were mounted in 0.1 M phosphate buffer on glass slides, dried and coverslipped with Dako fluorescence mounting medium.

**DAB—peroxidase staining for Ki67, DCX and NG2.** Free-floating sections were transferred from CPS into 0.01 M PBS and washed twice with 0.01 M PBS. Antigen retrieval was performed for 30 min at 40 °C using antigen recovery solution (2.6% sodium citrate, 3% sodium lauryl sulphate in dH$_2$O), followed by one wash each in dH$_2$O and 0.01 M PBS. Endogenous peroxidases were blocked through a 30 min incubation with 1% hydrogen peroxide (Merck Millipore). After thorough washes, the sections were incubated in 0.5% BSA,

0.05% saponin and 0.1% Triton X-100 in 0.01 M PBS for 1 h. Incubation with primary antibodies against Ki67 (rat anti-Ki67 1:8000, eBioscience, cat# 14-5698-82, RRID AB_10854564), DCX (rabbit anti-DCX, 1:8000, Cell Signaling Technology, cat# 4604, RRID AB_561007), or NG2 (rabbit anti-NG2, 1:2000, Merck Millipore, cat# AB5320, RRID AB_11213678), diluted in 0.5% BSA, 0.05% saponin and 0.1% Triton X-100 in 0.01 M PBS was performed for 48 h at 4 °C. Sections were washed thoroughly before incubation with biotinylated secondary antibodies (all 1:1000; goat anti-rat-biotin, Jackson ImmunoResearch, cat# 112-065-167, RRID AB_2338179 or goat anti-rabbit-biotin, Jackson ImmunoResearch, cat# 111-065-144, RRID AB_2337965) in 0.5% BSA, 0.05% saponin and 0.1% Triton X-100 in 0.01 M PBS was performed for 3 h at room temperature. After thorough washes, the Vectastain® Elite® ABC-HRP kit (Vector Laboratories, cat# PK-6100) was used by incubating the sections with Elite ABC complex solution for 1 h at room temperature. Following thorough washes, the sections were incubated for 5 min with 0.1 M sodium acetate buffer, followed by a 5 min incubation in freshly prepared nickel-3,3'-diaminobenzidine (DAB) solution (2% nickel(II) sulphate in 0.1 M sodium acetate buffer containing 0.025% DAB). Ki67$^+$, DCX$^+$ or NG2$^+$ cells were visualised through a 5 min incubation in nickel-DAB solution containing 0.015% hydrogen peroxide (Merck Millipore). The reaction was stopped through a 5 min wash step with 0.1 M sodium acetate buffer, after which the sections were washed two more times with 0.01 M PBS. They were then mounted in 0.1 M phosphate buffer on positively charged glass slides and dehydrated in ethanol and xylene, after which they were coverslipped with Ultramount (Dako Agilent).

### Cell counting and imaging

Cell counting was performed in a series of 40 μm sections, 240 μm apart, throughout the complete dentate gyrus area. Counting of fluorescently labelled cells was performed using a Zeiss microscope AxioImager Z1 with ApoTome attachment. Cells visualised based on DAB peroxidase staining were counted using an Olympus BX43 light microscope. All counting was performed blinded to the experimental groups.

### Volume measurements

Hippocampal volume and volume of the granular cell layer were measured in a complete one-in-six series of coronal section from mouse brain tissue of 8-week-old male PF4 knockout and wildtype animals. Images of DAPI-stained sections were acquired used a Zeiss microscope equipped with Apotome and the Zen Blue software. Areas of interest were determined using the Allen mouse brain reference atlas. Volume measurements were performed using ImageJ software (v2.1.0/153c).

### Branch tracing of DCX$^+$ cells

Images of the whole hippocampus were obtained from either 18-month-old female C57BL/6J running and standard-housed mice, 20-month-old female C57BL/6J mice receiving PF4 or saline and 12-month-old male PF4 knockout mice using a Yokogawa W1 spinning disk confocal microscope at ×200 magnification. DCX$^+$ cells were categorised by morphology, with only E and F cells (DCX$^+$ cells at postmitotic stages) that display defined dendritic branching[57] being selected for tracing analysis. Branch tracing was performed manually for each cell using Imaris software (Oxford Instruments, v10.0.0). All postmitotic DCX$^+$ cells from 5 mice per group were analysed, making a total of 32 cells from saline-treated mice and 36 cells from PF4-treated animals.

### Platelet depletion paradigm

Platelets were selectively depleted in 18-month-old triple-housed female C57BL/6J mice using rabbit anti-mouse antiplatelet serum (cat# WAK-AIA31440, Accurate Chemical & Scientific Corporation)[58]. The effective dose was determined for each batch of the serum, with a dose resulting in <50% reduction in platelet count compared to

standard-housed mice (equal to <500 × 10$^3$ platelets per microlitre blood) at 2 h post-injection being considered suitable for experiments. Here, this dose was determined as 1:2 in sterile 0.01 M PBS and administered at a volume of 150 μl. The mice were given intraperitoneal injections every second day for 30 days, making 16 injections in total. The control group received 150 μl of normal rabbit serum (cat# YNNRS, Accurate Chemical & Scientific Corporation). On day 3 of the experiment, the cages of mice in the running group were equipped with a running wheel that remained in place for the following 28 days of the experiment. The mice were perfused on the morning of day 29 and their brains collected for histology. Platelet counts were determined prior to perfusion using flow cytometry.

### Behavioural testing

The health of the aged mice was carefully monitored, their weight measured weekly, and their eyes examined for the development of cataracts, which would exclude them from behavioural testing. All mice were handled daily for at least 4 days prior to the start of the experiments.

### Novel object location task

Following PF4 or saline treatment of 20-month-old female C57BL/6J mice (i.v. for 24 days, every third day), the novel object location test was performed according to a standardised protocol[59]. During this task the mice were placed into a rectangular box (40 × 50 × 20 cm) in which they were free to explore two identical objects (~12 cm in height) for 10 min. A piece of black tape was placed along one edge of the box to assist with spatial navigation. The light intensity in the box was adjusted to measure 50 lux to avoid light-induced anxiety. Twenty-four hour later, one of the objects was shifted to a novel location. Movement and exploration behaviour of the mice in the box, as well as around both objects, were recorded through a ceiling-mounted camera. Between each animal, the box and objects were thoroughly cleaned with 70% ethanol. Analysis was performed using Ethovision XT software (Noldus, vXT14). Graphs represent data collected during the first minute of the test session.

### Contextual fear conditioning

Following PF4 or saline treatment (i.v. for 24 days, every third day), 20-month-old female C57BL/6J mice underwent a 10-min fear conditioning session, with four 0.8 mA shocks spaced 120 s apart, at 2, 4, 6 and 8 min of the task. The test was performed in individual soundproof operant chambers, which were scented with lemon essence. Between mice, each chamber was thoroughly cleaned with 70% ethanol and fresh lemon essence was applied to the chamber. Fear memory was assessed 24 h later by placing the mice into the same context for 5 min, during which no shocks were delivered. Freezing behaviour was recorded and analysed using Freeze Frame software (Actimetrics, v5).

### Active place avoidance test

Following systemic PF4 injections the mice (20-month-old female C57BL/6J or 20-month-old male and female DCX$^{DTR}$ mice) underwent the active place avoidance (APA) task. In this test, the mice were placed into a rotating cylindrical arena (Biosignal; 1 m in diameter) and had to learn to avoid a stationary 60° area (shock zone) using high contrast spatial cues that were evenly placed around the room. Detailed parameters of the testing equipment and setup have been described previously[21]. One day prior to testing, the mice underwent a 5 min habituation session in which the shock zone was inactive. Thereafter, 10 min testing sessions were performed each day for a total of 5 days. Between animals, the arena was thoroughly cleaned with 70% ethanol. Shocks were delivered at 0.5 mA, with an inter-shock latency of 1500 ms. A ceiling-mounted video camera paired with the tracking and shock computer was used to collect visual data, which were analysed using APA analysis software (Tracker v236) from Biosignal.

Improvement during the APA task was calculated for each animal and represented as percentage change based on the number of shocks received on the first vs. the last day of testing.

## Isolation of washed platelets for proteomic analysis

Eight-week-old or 18-month-old female C57BL/6J mice were either standard housed or housed in cages with a running wheel for 4 or 28 days. The animals were then deeply anaesthetised with 100 mg/kg ketamine/20 mg/kg xylazine in 0.9% NaCl, after which whole blood was collected from the vena cava into ACD buffer (85 mM trisodium citrate, 71 mM citric acid, 111 mM D-glucose). The suspension was mixed with washing buffer (0.001 M EGTA, 12 mM NaHCO$_3$, 13 mM BSA, 9.98% modified Tyrode's calcium-free buffer pH 7.2 (134 mM NaCl, 3 mM KCl, 0.3 mM NaH$_2$PO$_4$*H$_2$O, 5 mM HEPES, 5 mM D-glucose, 2 mM MgCl$_2$ in dH$_2$O)) and centrifuged at $180 \times g$ for 10 min at room temperature without brake. The supernatant (platelet-rich plasma) was collected and mixed with washing buffer containing prostaglandin E1 (PGE$_1$; final concentration 0.25 μmol/l) followed by centrifugation at $1250 \times g$ for 10 min at room temperature without brake. The pelleted platelets were washed again with washing buffer containing PGE1 (final concentration 0.25 μmol/l) and centrifuged at $1250 \times g$ for 10 min at room temperature without brake. The pelleted platelets were then washed with resuspension buffer (12 mM NaHCO$_3$, 13 mM BSA, 9.98% modified Tyrode's calcium-free buffer pH 7.2 in dH$_2$O) and centrifuged $1250 \times g$ for 10 min at room temperature without brake. The supernatant was removed to leave a volume of 115 μl, in which the pelleted platelets were resuspended. Aliquots were prepared for protein estimation and proteomic analysis and stored at −80 °C.

## TMT quantitative proteomic analysis of platelet lysate

Protein estimation for each sample was performed using the Pierce BCA protein assay kit to ensure that each sample contained at least 300 μg of protein. Platelet lysate was prepared by three repeated freeze-thaw cycles and the frozen samples were sent on dry ice to The Australian Proteome Analysis Facility (APAF), which performed the proteomics analysis. For this, the samples were digested using commercially procured S-Traps (Protifi) and equal total peptide quantities were prepared for mass spectrometry using Tandem Mass Tag (TMT; Thermo Scientific) reagent labelling. The TMT-labelled peptides were fractionated using Offline Basic pH reversed-phase fractionation, dried by vacuum centrifugation, and reconstituted in 0.1% formic acid for liquid chromatography with tandem mass spectrometry (LC-MS/MS) analysis, which was performed using the UltiMate 3000 nanoLC system (Thermo Fisher Scientific) and a Q-Exactive HF-X mass spectrometer (Thermo Fisher Scientific). It is of note that while this traditional approach identifies a large number of differentially expressed proteins, information about their structural changes is not detected. Raw data files were processed using the Proteome Discoverer software (Thermo Scientific, v2.1.0.81). The data were searched using the search engines SequestHT and Mascot against a sequence database for *Mus musculus*.

## Analysis of proteomics data

The proteomic analyses of mouse platelet lysate resulted in the identification and quantification of 2625 (aged mice; 6 running and 7 standard-housed) and 2639 (young mice; 5 mice per group) proteins across all samples (protein, peptide, peptide-spectrum matches; false discovery rate <1%). Following a pairwise relative-abundance comparison across all sample groups using two-tailed $t$-tests, proteins with a $p$ value smaller than 0.05 and a fold change greater than ±1.2 were determined as proteins with differential protein levels between groups (Supplementary Data 4 and 5). Overlapping proteins were determined and visualised in a Venn diagram using Venny 2.1.0[60]. Gene ontology (GO)[61,62] enrichment analyses were performed with the corresponding genes of protein lists using clusterProfiler 4.0[63] in April 2022, to identify significantly enriched biological processes and molecular functions following exercise. Graphs

show ranked candidate biological processes determined by Benjamini–Hochberg-adjusted $p$ values. A complete list of GO results is provided in Supplementary Data 6 for young mice and Supplementary Data 7 for aged mice. The heatmap of protein changes was created using Morpheus (https://software.broadinstitute.org/morpheus/).

## RNA sequencing of adult neural stem cells and dentate gyrus cells

Primary cells were isolated from the dentate gyrus of 8-week-old female C57BL/6J mice as described earlier[13]. The dentate gyri of two mice were pooled for each treatment condition or staining control. The cells were collected in 1 ml growth medium consisting of neural basal medium containing 2% B-27® supplement (50X), 1% penicillin/streptomycin (10,000 U/ml) and 1% GlutaMAX™ (100X), transferred to a 24-well plate and treated with either 100 ng/ml recombinant PF4 protein or saline, then incubated at 37 °C and 5% CO$_2$ for 1.5 h. After this, the cells were stained with EGF complexed to Alexa Fluor 647-streptavidin (EGF-647; Thermo Fisher cat# E35351) for 30 min to label proliferating neural stem cells. EGF-647 was used at 0.5 μg/ml, as determined by a titration study, which was performed prior to the experiment. The cells were then transferred to a 15 ml tube and washed with 0.01 M PBS followed by centrifugation at $300 \times g$ for 5 min. The pelleted cells were resuspended in 600 μl 0.01 M PBS containing SUPERase•In™ RNase Inhibitor (1:1000 Invitrogen, cat# AM2694), transferred into a 5 ml flow cytometry tube, in which propidium iodine (PI, final concentration 1 μg/ml) was added to distinguish alive from dead cells during flow cytometry. EGF$^+$ neural stem cells and EGF$^-$ niche cells were collected using a BD FACSAria cell sorter with FACS-Diva software (v9.0) into collection tubes containing 50 μl 0.01 M PBS supplemented with SUPERase•In™ RNase Inhibitor (1:1000; Invitrogen, cat# AM2694). Fluorescence minus one controls (PI only and EGF-647 only) were used to determine background and gate the respective populations. RNA isolation with DNase treatment was performed immediately after cell collection using an Qiagen RNeasy micro kit. Quality control of the isolated RNA was preformed using the Agilent 2100 BioAnalyzer and Perkin Elmer LabChip GX. cDNA was prepared using the SMART-Seq v4 Ultra Low Input RNA kit (Takaro Bio) followed by library preparation with the Illumina Nextera XT DNA library preparation kit. Six samples from each condition, EGF$^+$ adult neural stem cells and EGF$^-$ dentate gyrus cells that were treated with either PF4 or saline, were sequenced using the Illumina Nova Seq 6000 S1 system.

## Analysis of sequencing data

Raw read files were processed with Trim Galore v0.6.10 (https://github.com/FelixKrueger/TrimGalore) to perform quality control and remove adaptors. Processed reads were pseudo-aligned using Salmon v1.10.0[64] to a prebuilt MM10 genome index (http://refgenomes.databio.org/v3/genomes/splash/0f10d83b1050c08dd53189986f60970b92a315aa7a16a6f1). These data were imported into an R v4.2.2 environment using the tximport v1.26.1[65] and TxDb.Mmusculus.UCSC.mm10.ensGene v3.4.0[66] libraries. Differential gene expression analysis was performed using DESeq2 v1.38.3[67], which models RNA-sequencing counts through a negative binomial distribution model, followed by a Wald test and Benjamini–Hochberg correction for multiple testing. EnhancedVolcano v1.16.0[68] produced volcano plots, with shrunken log fold change values calculated using "ashr" as an estimator. The Pheatmap v1.0.12 package[69] was used to generate heatmaps of log$_2$-normalised expression values where adjusted differentially expressed contrast $p$ values were <0.05. Euler plots[70] of up- and downregulated genes either common or unique to specified treatments were generated with eulerr v7.0.0[71]. Functional enrichment analyses, including GO[61,62] enrichment, were performed using g:Profiler (v e108_eg55_p17_0254fbf) in February and March 2023, with g:SCS multiple testing correction method applying significance threshold of 0.05[72]. A complete list of GO results is provided in Supplementary Data 3.

## Statistical analysis

Analyses other than for the sequencing or proteomics data were done using Graph Pad Prism v9. All data in bar graphs are represented as mean ± SEM. Comparisons of two groups were performed using unpaired, two-tailed Student's $t$ tests with assumed equal variances or Mann–Whitney $U$ tests, as appropriate. For comparisons of more than two groups, one-way or repeated measures two-way ANOVAs with post hoc Dunnett or Sidak comparison were performed, as appropriate. Details of all test results are listed in Supplementary Data 8.

## Reporting summary

Further information on research design is available in the Nature Portfolio Reporting Summary linked to this article.

## Data availability

Raw sequencing data that support the findings of this study have been made openly available in the National Center for Biotechnology Information repository at https://www.ncbi.nlm.nih.gov/bioproject/ under accession code PRJNA938247. Processed reads from our sequencing data were pseudo-aligned using Salmon v1.10.0 to a pre-built mm10 genome index (http://refgenomes.databio.org/v3/genomes/splash/0f10d83b1050c08dd53189986f60970b92a315aa7a16a6f1). The mass spectrometry proteomics data have been deposited to the ProteomeXchange Consortium[73] via the PRIDE[74] partner repository with the dataset identifiers PXD041700 (young mice) and PXD041701 (aged mice), and can be accessed at https://www.ebi.ac.uk/pride/. Source data are provided with this paper.

## Code availability

All R code used for the analyses is available at https://github.com/Walker-Neurogenesis/PF4.

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

## Acknowledgements

This project was partly funded by The Clem Jones Centre for Ageing Dementia Research, National Institutes of Health (R01AG077816; T.L.W. and S.A.V.) and received philanthropic support from The Brazil Family Program for Neurology and The Donald & Joan Wilson Foundation Ltd. O.L. received a Walter Benjamin Postdoctoral Fellowship from the

Deutsche Forschungsgemeinschaft (DFG, German Research Foundation, project number 668329) and postdoctoral fellowship from the Deutscher Akademischer Auslandsdienst (DAAD; German Academic Exchange Service). V.A. was supported by an Australian Research Council Future Fellowship (FT220100485). X.L.H.Y. received a Research Training Program Scholarship from the Australian Government and the University of Queensland, as well as the Ian Lindenmayer PhD Top-up Scholarship. This study used the Queensland Brain Institute's Animal Behavioural, Histology, Advanced Microscopy and Flow Cytometry facilities and NCRIS-enabled Australian Proteome Analysis Facility (APAF) infrastructure, and the authors acknowledge the contribution of all facility staff. The authors thank the animal house staff for their assistance with animal maintenance and care, and Rowan Tweedale for her comments on the paper.

## Author contributions

O.L. and T.L.W. designed the study and wrote the manuscript. O.L., D.B., J.W., V.A., X.L.H.Y., A.B.S., G.B. and T.L.W. performed experiments. O.L., S.J.F., J.W., X.L.H.Y. and N.M. analysed the data. D.G.B., P.F.B. and S.A.V. provided animal models. O.L., P.F.B., S.A.V. and T.L.W. acquired funding.

## Competing interests

The authors declare no competing interests.
