## [Peer Review File · Nature Communications]

Platelet-derived exerkine CXCL4/platelet factor 4 rejuvenates hippocampal neurogenesis and restores cognitive function in aged miceREVIEWER COMMENTS

Reviewer #1 (Remarks to the Author):

This is an interesting manuscript presenting data showing that a platelet derived protein, platelet factor 4 (PF4), can stimulate adult neurogenesis in the dentate gyrus of old mice and that the increase in adult neurogenesis contributes to restored cognitive function. The findings are present a novel view of platelet function and are potentially significant clinically. I have the following suggestions to make the findings more convincing and to further expand the impact of the study:

- 1) Running distance data are only provided for the PF4 knockout mice but not for the mice treated with antiplatelet serum or for the DCX-DTR study. It is important to include these data because differences in running distance as a result of treatment could explain some of the data.
- 2) More evidence that PF4 effects are specific to adult neurogenesis in the dentate gyrus should be included. Evidence of no change after treatment in mature neurons and synapses would be helpful (in populations that do not undergo adult neurogenesis, like hippocampal pyramidal cells). In addition, looking at whether cell proliferation in the SVZ is affected, as well as whether oligodendrocyte progenitor cells (which have been linked to enhanced cognition) are affected by treatment would be helpful.
- 3) More details should be provided about the transgenic mice. Did the PF4 knockout mice have any phenotypic differences? Any differences in other hippocampal measures? Did the DCX-DTR mice show any hippocampal changes other than in new neurons after DTR treatment? Evidence to the contrary would help to establish that these additional effects are not responsible for the changes in behavior.
- 4) The authors mention "small but not significant effects" and also begin the paper with a change that has a p value of .05 (Figure 1C). These effects should be explored further - maybe the study is underpowered? At any rate, it is not convincing to report effects that do not reach statistical significance.
- 5) It is confusing that the young mice do not show cell proliferation effects after PF4 treatment but the old mice do. What is the reason for this difference? This should be discussed in the paper.
- 6) Why were the young mice singly housed but the old mice were not? There is evidence that social isolation impacts adult neurogenesis and cognition. How might the housing affect the results - this should be discussed in the paper.
- 7) Were the blood samples at different time points all taken from the same mice? If so, was there any evidence of changes over time that might relate to chronic stress?
- 8) Only one cognitive task is included (active place avoidance). In order to consider whether these effects are more generalizable, additional hippocampus-dependent effects should be included.
- 9) There is a big gap between the histology and behavior findings. Providing some electrophysiology data would be helpful to increase our understanding of how PF4-induced adult neurogenesis leads to restored cognitive function in old mice.

Reviewer #2 (Remarks to the Author):

The rejuvenating benefits of exercise on old mice are identified by the authors of the current study as requiring platelets as mediators.

They demonstrate that the exerkin PF4 produced by platelets is both adequate and essential for mediating this mechanism.

Furthermore, by promoting adult hippocampal neurogenesis and regaining cognitive function in the aging brain, they show that systemic PF4 administration can imitate the rejuvenating benefits of exercise.

The authors additionally demonstrate that neurogenesis is essential for PF4-mediated cognitive regeneration using an aged transgenic mice model of neurogenesis ablation. These data provide strong evidence that PF4 might be a suitable target to improve age-related cognitive decline. The text is well written and the appropriate methods were chosen. In some cases the statements were overstretched; e.g. `we were the first to show` or data show an increase without reaching significance (see below). In general, the references were all covered. The strength of the manuscript is clearly the timely topic and the option for the reader to further understand in more detail how exercise strengthens cognition.

In order to improve the quality of the manuscript, some aspects have to be addressed.

Major:

a) Fig. 1 c there is no significance level reached (* $p < 0.05$) so it cannot be stated that the number of DCX pos. cells increased.

b) It is puzzling that the exercise-induced increase in precursor cell proliferation was absent in mice lacking PF4. On the other hand, application of PF4 had no impact on the proliferation rate. Can the authors comment on this issue.

c) Instead of just showing that Ki67 pos cells increase, it would be interesting to see which cells are proliferating (e.g. Tbr2+, Sox2+, DCX+)

d) The authors claim that "systemic PF4 treatment of aged mice reversed this cognitive deficit. This was observed as an overall improvement in the performance". When looking at Fig. 5e, this statement is not correct because there is no overall improvement.

e) It would be interesting to see whether the morphology of DCX+ cells is changed after long-term running or whether only the cell number increases.

f) In the discussion the authors point out that: "Here we provide the first evidence that platelets also release exerkines". This statement is not correct. See: Cai et al., 2020 (DOI: 10.3390/antib9040052)

g) The manuscript would be improved if authors could show whether PF4 is taken up by the cells. It was shown for monocytes but it is absolutely unknown whether neuronal precursor cells can actually incorporate PF4 as well.

Minor:

a) Fig 3a-c; the authors show only the quantification of immunohistochemical experiments, adding the original images would be necessary.

b) Original images depicting DCX+ cells are beneficial here.

Reviewer #3 (Remarks to the Author):

These data show that the cognition-enhancing effects of endurance training of aged mice are mediated by platelets and can be replicated by the systemic administration of platelet-released exerkine PF4 in a hippocampal neurogenesis-dependent manner. This research approach is very interesting because cognition-enhancing effects may be achieved by platelets, which are very high concentrated in the blood circulation. Especially because it has been described that the platelet secretome contains many regeneration-promoting growth factors.

However, the physiological mode of action of how PF-4 is supposed to induce both its release from platelets and direct signal transduction of neurogenesis in hippocampus remains largely unclear in this work. At this point, it would be also important to study how PF4 would mediate this neurogenesis-promoting effect in a neuronal cell culture model. Instead, platelet depletion experiments and platelet proteomics analyzes using TMT shotgun technology were carried out in various mouse models, which are of lesser importance in clarifying this PF-4 neurogenesis-promoting mechanism of action.

The authors describe, as if it would be new research findings, that increased platelet activation occurs shortly after exercise. It has long been known that platelets are activated by exercise and during this process release numerous proteins known for platelet activation, such as e.g. B. sCD62P.

Although this work also describes that more weekly endurance training and the persistently recurring platelet activation and associated increased release of PF4 should have a cognitive-promoting effect. In fact, it has been described that

endurance exercise, which counteracts vascular disease (PMID: 26557653) as well as Alzheimer's disease, reduces basal platelet activation, platelet reactivity (also PF-4 release should be reduced in circulation) and therefore also has an antithrombotic effect. All of these previous research data in the literature were insufficiently discussed in this work.

Why these effects cannot not also be mediated for example by BDNF for example?

If they hypothesize that PF4 promotes neurogenesis, why did the researchers perform platelet proteomics? Interestingly, the platelet proteome studies showed that not more PF4 level could be detected in mice after doing endurance training. However this PF4 data would be important to be shown in the main figures! At this point, it would also have been more important to investigate whether more PF-4 could be detected in the plasma of the exercising mice. Or if the PF4-release is changed in platelets of trained mice.

The platelet proteome data of this work showed that the PF4 levels in the platelet proteome of young and old mice were not changed after

endurance training as well as other exercise-dependent exerkines like cathepsin B and clusterin.... It should also be additionally noted as further limitations of this study work that proteomic shotgun analysis (bottom-up proteomics) only measures the levels of canonical proteins and cannot distinguish between functional proteoforms, in contrast to top-down proteomics, which also detect functional proteoform differences can.

Inactive and active protein such as cathepsin B. (Regulation of cathepsin B functionality is regulated by proteolytic cleavage of cathepsin B).

The researchers also found that tropomyosin 1 levels increased after exercise in both old and young mice. Interestingly, it has previously

been found that tropomyosin is elevated both in platelets from female Alzheimer's patients and in brains from Alzheimer's cases. But here

it is important to know that tropomyosin 1 in platelets has numerous proteoforms and only its high molecular weight protein species are

increased in platelets of Alzheimer's patients

If the researchers want to use their platelet proteome data, it can be helpful to correlate neurogenesis data with the training level of the

young and old mice to find potentially significant and hopefully knowledge-enhancing correlations. These results could indicate which

protein in the platelet proteome might be significantly involved in neurogenesis.

with best regards,
Maria Zellner

REVIEWER COMMENTS

Reviewer #1 (Remarks to the Author):

This is an interesting manuscript presenting data showing that a platelet derived protein, platelet factor 4 (PF4), can stimulate adult neurogenesis in the dentate gyrus of old mice and that the increase in adult neurogenesis contributes to restored cognitive function. The findings are present a novel view of platelet function and are potentially significant clinically. I have the following suggestions to make the findings more convincing and to further expand the impact of the study:

1) Running distance data are only provided for the PF4 knockout mice but not for the mice treated with antiplatelet serum or for the DCX-DTR study. It is important to include these data because differences in running distance as a result of treatment could explain some of the data.

Mice in the antiplatelet serum study were housed in groups of three for the duration of the experiment (31 days) to ensure our results and running behaviour was not influenced by stress that could be induced by single housing. The running behaviour of these mice was monitored using a ceiling mounted camera. This revealed that the mice took turns using the running wheels and sometimes also shared the running wheel while exercising. We have now performed a detailed analysis of the time that these mice spent using the running wheels (running time per night per cage) and found no differences between running mice that received control serum and antiplatelet serum. We have included these data in the revised manuscript (p. 13, line 294-297 and Supplementary fig. S8).

The DCX^{DTR} study was performed to determine whether the PF4-mediated improvement in learning and memory in aged animals was dependent on increased adult hippocampal neurogenesis. To test this, 20-month-old DCX^{DTR} mice received PF4 or saline every third day for 24 days. Towards the end of the treatment paradigm, a subset of PF4 and saline-treated mice received diphtheria toxin (DT) injections to ablate ongoing neurogenesis, before the active place avoidance task was performed to test their spatial learning and memory ability. This experiment did not include running.

2) More evidence that PF4 effects are specific to adult neurogenesis in the dentate gyrus should be included. Evidence of no change after treatment in mature neurons and synapses would be helpful (in populations that do not undergo adult neurogenesis, like hippocampal pyramidal cells). In addition, looking at whether cell proliferation in the SVZ is affected, as well as whether oligodendrocyte progenitor cells (which have been linked to enhanced cognition) are affected by treatment would be helpful.

To address this question, we first determined whether PF4 treatment affects synaptic composition by examining the expression of PSD-95 (a postsynaptic marker) and synaptophysin (a presynaptic marker) in the dentate gyrus of PF4-treated mice. We observed no differences in protein expression in the crude synaptosomal fraction (P2) or in the total homogenate of dentate gyrus tissue following PF4 treatment. Our data suggest that PF4 treatment is unlikely to alter synapse density in the dentate gyrus. To determine whether PF4 treatment alters the neurite complexity of mature neurons we treated mature hippocampal neurons with saline or PF4 and performed Sholl analysis. Again, we observed no differences between the treatment groups. A paragraph summarising these data has been included in the main text of the manuscript (p. 8-9, lines 186-200), with the results being presented in Supplementary fig. 4.

In addition to investigating the effects of PF4 treatment on neural precursor cell proliferation in the dentate gyrus, we have now also determined whether this treatment affects subventricular zone (SVZ) precursor proliferation. We found that SVZ proliferation remained unaffected in both aged and young animals receiving PF4 treatment. Moreover, while the numbers of proliferating cells and immature neurons in the dentate gyrus were decreased in the dentate gyrus of PF4 knockout mice, SVZ proliferation was not affected. This is in accordance with our previous data showing that PF4 treatment *in vitro* only affects primary cells derived from the dentate gyrus, but not the SVZ (Leiter et al., 2019, *Stem Cell Reports*). These data have now been included in the revised manuscript for PF4 KO mice on p.6, lines 134-140 and for aged PF4-treated animals on p.14, lines 324-327. Corresponding graphs and representative images of the staining are shown in Supplementary fig. 2d, e (PF4 knockout mice), as well as in Supplementary fig. 9c, d (aged animals).

To address whether oligodendrocyte progenitor cells are affected by PF4 treatment, we have quantified the number of NG2⁺ cells in the dentate gyrus and subgranular zone of 20-month-

old PF4-treated mice, in which we observed an increased number of Ki67⁺ proliferating cells. This revealed no difference in the number of NG2⁺ cells between saline and PF4-treated animals, suggesting that oligodendrocyte progenitor cells in the dentate gyrus are not responsive to the treatment. These data are now included in the manuscript on p. 13, lines 309-313, and Supplementary fig. 9a, b.

3) More details should be provided about the transgenic mice. Did the PF4 knockout mice have any phenotypic differences? Any differences in other hippocampal measures?

We have now included this information in the main text (p. 6, lines 126-130), with a reference to the original publication (Eslin et al., 2004, *Blood*), which reads:

“For this analysis we used 8-week-old PF4 knockout mice, in which platelets are devoid of PF4 mRNA or protein¹⁴. Although these mice exhibit a higher platelet count compared to wildtype littermates, other hematologic parameters, such as platelet function, as well as their overall appearance and anatomy, weight, survival and fertility, remain unaffected¹⁴.”

We have now also characterised the hippocampal and dentate gyrus volume this mouse model. We found no difference between PF4 knockout mice and their wildtype littermates. As addressed in question 2), we also quantified neural precursor cell proliferation in the SVZ, with no differences between knockout mice and wildtype littermates. These data are included in the revised manuscript on p. 6, lines 132-138, and Supplementary fig. 2a – c.

Did the DCX-DTR mice show any hippocampal changes other than in new neurons after DTR treatment? Evidence to the contrary would help to establish that these additional effects are not responsible for the changes in behavior.

This mouse model has been extensively characterised (Vukovic et al., 2013, *Journal of Neuroscience*). Specifically, in these mice, treatment with human diphtheria toxin (DT) exclusively affects DCX⁺ cells in the hippocampus, whereas the number of proliferating Ki67⁺ cells and the number of hippocampal neurospheres derived from the dentate gyrus of DT-injected animals remain unaffected. Moreover, DT treatment does not induce gliosis or microglial activation. Therefore, the observed changes in the behaviour of DT-treated mice can be attributed the reduction in the number of DCX⁺ cells.

4) The authors mention "small but not significant effects" and also begin the paper with a change that has a p value of .05 (Figure 1C). These effects should be explored further - maybe the study is underpowered? At any rate, it is not convincing to report effects that do not reach statistical significance.

In the above-mentioned experiment, 8-week-old C56BL/6J mice (n = 10 mice per group) received systemic injections of either saline or PF4 every second day for one week. We have now repeated the experiment with an additional n = 5 mice per group. The combined data (n = 15 mice per group) revealed a significant increase ($p = 0.03$) in the number of immature neurons in PF4-treated mice compared to saline-treated controls. This has been updated in the manuscript, including the graph shown in figure 1c, and on p. 4 lines 76-79 and now reads:

“Similar to the effect observed following systemic delivery of plasma isolated from exercised aged mice ⁶, systemically delivered PF4 increased the number of doublecortin⁺ (DCX⁺) immature neurons in the dentate gyrus without affecting neural precursor cell proliferation (Ki67⁺ cells) in young adult mice (Fig 1b, c).”

5) It is confusing that the young mice do not show cell proliferation effects after PF4 treatment but the old mice do. What is the reason for this difference? This should be discussed in the paper.

We agree that this is a somewhat contradictory finding. One possible explanation for the difference observed between the young and old mice is that aged mice exhibit extremely low levels of baseline proliferation. Therefore, even very small increases in the absolute number of proliferating cells following PF4 treatment have a significant effect on the total cell count. Such effects could, however, be masked in 8-week-old animals, in which the levels of baseline proliferation are approximately 40 times higher than in 20-month-old mice. We have added this to the discussion on p. 19-20, lines 461-468.

6) Why were the young mice singly housed but the old mice were not? There is evidence that social isolation impacts adult neurogenesis and cognition. How might the housing affect the results - this should be discussed in the paper.

Although most of the experiments were performed with mice that were group-housed, experiments with young mice, which underwent a short-term running paradigm, were performed with single-housed animals in order to acquire accurate wheel usage information. This included the PF4 knockout mouse study and mice housed 4 days in cages with or without running wheels for the collection of platelets for proteomic analysis. Results from our previous work confirm that serum corticosterone levels in 8-week-old female C57BL/6J mice, which are single-housed for a short period of time, including 4 days of exercise, do not have increased serum corticosterone levels (Overall, Walker et al., 2013, *PloS One*), suggesting that this paradigm does not induce stress in the animals. As the experiments with aged mice followed a longer timeline (28 days for running experiments), all aged running animals were housed in groups of three, which allowed them to share the running wheels and avoided adverse effects on the behaviour of the mice which could be caused by long-term social isolation. All aged mice in the PF4-treatment neurogenesis studies and all mice used in behaviour experiments were group housed.

The reasons for the single or group housing in our exercise studies are now specified in the methods sections (p. 22, lines 536-540) which reads:

“Young mice (short-term running; 8 weeks old) were single housed for the running duration in order to acquire accurate wheel usage information, whereas aged mice (longer periods; 18 months old) were housed three per cage to avoid adverse effects on the behaviour of the mice which could be caused by long-term social isolation.”

7) Were the blood samples at different time points all taken from the same mice? If so, was there any evidence of changes over time that might relate to chronic stress?

Yes, the platelet activation study on 18-month-old C57BL/6J standard-housed and running mice was a longitudinal experiment in which the blood samples were taken from the same mice on each testing day. This has now been clarified in the methods section (p. 27, lines 644-646). Prior to the experiment, the mice were handled daily for one week and habituated to the experimental room to reduce stress. We have now also measured corticosterone levels in the serum of these mice at the beginning (day 1) and end (day 28) of the exercise paradigm. This analysis revealed no differences between any of the groups. This result is included in the manuscript on p. 12, lines 276-278, and Supplementary fig. 6b.

8) Only one cognitive task is included (active place avoidance). In order to consider whether these effects are more generalizable, additional hippocampus-dependent effects should be included.

We have now performed additional hippocampus-dependent tasks, following our 24-day PF4 treatment protocol. Two separate cohorts of 20-month-old C57BL/6J mice were tested in the novel object location task, followed by contextual fear conditioning, with the combined data totalling 22 mice in the saline-treated group and $n = 24$ mice receiving PF4. In both tasks, the PF4-treated mice show significant improvements in their performance, as detailed in the added paragraph on p. 15-16, lines 348-372, and Fig. 6 and Supplementary fig. 11.

9) There is a big gap between the histology and behavior findings. Providing some electrophysiology data would be helpful to increase our understanding of how PF4-induced adult neurogenesis leads to restored cognitive function in old mice.

In our experiment using the DCX^{DTR} mouse model, we showed that the PF4-mediated improvement in cognition is abolished when neurogenesis is ablated, thereby directly linking PF4-mediated increases in adult hippocampal neurogenesis detected by histology to improvement in hippocampus-associated learning and memory. In our recent publication, which used the same mouse model, we showed a similar link between exercise-mediated improvement in cognition and new neuron production (Blackmore et al., 2021, *iScience*). Although beyond the scope of our study, we are currently co-ordinating the publication of our manuscript with a complementary manuscript by colleague Associate Professor Dena Dubal. In their study (currently under review at *Nature*; manuscript # 2021-07-11661B) they show that PF4 treatment of isolated hippocampal slices increases long-term potentiation in the hippocampus.

Reviewer #2 (Remarks to the Author):

The rejuvenating benefits of exercise on old mice are identified by the authors of the current study as requiring platelets as mediators. They demonstrate that the exerkin PF4 produced by platelets is both adequate and essential for mediating this mechanism. Furthermore, by promoting adult hippocampal neurogenesis and regaining cognitive function in the aging brain, they show that systemic PF4 administration can imitate the rejuvenating benefits of exercise. The authors additionally demonstrate that neurogenesis is essential for PF4-mediated cognitive regeneration using an aged transgenic mice model of neurogenesis ablation. These data provide strong evidence that PF4 might be a suitable target to improve age-related cognitive decline. The text is well written and the appropriate methods were chosen. In some cases the statements were overstretched; e.g. we were the first to show or data show an increase without reaching significance (see below). In general, the references were all covered. The strength of the manuscript is clearly the timely topic and the option for the reader to further understand in more detail how exercise strengthens cognition.

In order to improve the quality of the manuscript, some aspects have to be addressed.

Major:

a) Fig. 1 c there is no significance level reached (* $p < 0.05$) so it cannot be stated that the number of DCX pos. cells increased.

Please see the response to reviewer 1, point 4.

b) It is puzzling that the exercise-induced increase in precursor cell proliferation was absent in mice lacking PF4. On the other hand, application of PF4 had no impact on the proliferation rate. Can the authors comment on this issue.

We were not surprised that the exercise-mediated increase in proliferation was absent in PF4 knockout mice. We believe that this response is mediated by activated platelet-released PF4, which is absent in the KO mice. Moreover, PF4 could be directly involved in the regulation of platelet activation following exercise, a response which will be dampened in the PF4 KO mice.

We agree with the reviewer that it is somewhat contradictory that PF4 treatment does not increase proliferation in the young hippocampus. We do, however, observe a significant increase in proliferation in the aged PF4-treated mice. For a possible explanation of the difference observed between the young and aged mice, please refer to our comment to reviewer 1, point 5. We would like to highlight that our finding in young mice agrees with our previous work, in which we observed the same effect of PF4 (delivery directly into the hippocampus of 8-week-old C57BL/6J mice using mini osmotic pumps) on DCX⁺ cells without affecting the number of Ki67⁺ cells (Leiter et al., 2019, *Stem Cell Reports*). To confirm our *in vivo* finding of an effect of PF4 on neurogenesis, but not proliferation, we have now performed additional *in vitro* experiments to determine the effects of PF4 on proliferation (using BrdU labelling; n = 6 individual experiments), cell cycle (using clickIT[®] EdU technology; n = 7 individual experiments) and neuronal lineage differentiation (using the differentiation assay; n = 6 individual experiments). The results of these assays support our findings of a pro-neurogenic, rather than a pro-proliferative effect of PF4 on neural precursor cells. A paragraph detailing the results from the *in vitro* studies has been included in the revised manuscript on p. 5-6 lines 95-122, and Supplementary fig. 1.

c) Instead of just showing that Ki67 pos cells increase, it would be interesting to see which cells are proliferating (e.g. Tbr2⁺, Sox2⁺, DCX⁺)

We thank the reviewer for this suggestion. We have now performed further phenotyping analysis of the proliferating cells in the aged mice following PF4 and saline treatment using two different antibody combinations: Ki67, Sox2 and Tbr2 to label the proliferative neural precursor cells at early stages of neuronal development and Ki67 in combination with DCX to label the more mature proliferating neural precursor cells. This revealed that most of the proliferating cells (combined approximately 90% in both treatment groups) were Sox2⁺Tbr2⁻ or Sox2⁺Tbr2⁺, confirming that they belonged to the neuronal lineage. Very few proliferating cells were Sox2⁻Tbr2⁺ (1.3% ± 1.9% in saline-treated mice vs. 3.0 ± 2.4% in PF4-treated animals). We also observed no significant differences in the relative proportion of the cells expressing each marker between PF4-treated and control mice. However, when we compared the proportion of proliferating neural precursor cells that were at a later developmental stage, we observed an increase in Ki67⁺ DCX⁺ immature neurons following PF4 treatment, suggesting an effect on this more mature population. This is in accordance with the increase in immature neurons that we observed in the dentate gyrus of aged sedentary mice following PF4

supplementation. We also investigated whether the increase in proliferation could be due to changes in the number of NG2⁺ glia, which are known for their highly proliferative phenotype. However, we found that the number of NG2⁺ cells in the dentate gyrus and subventricular zone remained unaffected by PF4 treatment. A paragraph summarizing these data has now been included in the revised manuscript on p. 13-14, lines 308-324, and figure 5d and e, as well as Supplementary figure 9.

d) The authors claim that "systemic PF4 treatment of aged mice reversed this cognitive deficit. This was observed as an overall improvement in the performance". When looking at Fig. 5e, this statement is not correct because there is no overall improvement.

We thank the reviewer for pointing this out. We agree that this statement was confusing and have now removed the graph in Fig. 5e (now Fig. 6) which showed the % improvement of the mice in entering the shock zone during the active place avoidance (APA) task. However, other measures of the APA task confirm that the 20-month-old PF4-treated mice perform better than saline-treated control animals. To confirm the findings of the APA test, we have now also performed two additional hippocampus-dependent tasks following our 24-day PF4 treatment protocol: the novel object location task and contextual fear conditioning (n = 22 mice in the saline-treated group and n = 24 mice receiving PF4). In both tasks, the PF4-treated mice show significant improvements in their performance, confirming our results from the APA test. Data from both the novel object location and fear conditioning tasks have been added to the manuscript on p. 15-16, lines 348-372, and Fig. 6 and Supplementary fig. 11.

e) It would be interesting to see whether the morphology of DCX⁺ cells is changed after long-term running or whether only the cell number increases.

This is an interesting idea, which we have now addressed by investigating the dendritic architecture of DCX⁺ cells in the cohort of aged mice undergoing 28 days of exercise (n = 66 cells from standard-housed mice and n = 71 cells from running mice). We found that, similar to the immature neurons of PF4-treated mice, those of exercising animals exhibited a significant increase in the total dendritic length. However, although PF4 promoted the elongation of dendrites, the increase in dendritic length in aged exercising mice could be attributed to increases in branch complexity. These data have been added to the manuscript on p. 14-15 lines 341-345, and Supplementary fig. 10.

f) In the discussion the authors point out that: "Here we provide the first evidence that platelets also release exerkinines". This statement is not correct. See: Cai et al., 2020 (DOI: 10.3390/antib9040052)

The paper to which the reviewer is referring does not investigate exercise and therefore does not identify PF4 as an “exerkine”. We agree that it is well-known that PF4 is released by activated platelets, and also following exercise. Our study is the first to define PF4 as a platelet-released exerkine with pro-cognitive effects, confirming an important role of platelets in the regulation of brain function. We have clarified this in the discussion (p. 18, lines 438-440), which now reads:

“Our results provide the first evidence that platelets release exerkinines, including PF4, that increase adult neurogenesis and rejuvenate cognitive function, thereby cementing their important role in the regulation of brain function.”

g) The manuscript would be improved if authors could show whether PF4 is taken up by the cells. It was shown for monocytes but it is absolutely unknown whether neuronal precursor cells can actually incorporate PF4 as well.

We have now performed additional experiments to investigate whether PF4 can be taken up by adult neural precursor cells, using an established adherent hippocampal monolayer culture model (Bernas, Leiter et al., 2017, *Bio-protocol*). To do this we labelled recombinant PF4 protein with a fluorescent dye using Lightning Link[®] technology. This enables specific labelling of the protein and includes a quenching step to eliminate any remaining free label. Culturing of the neural precursor cells in the presence of labelled PF4 revealed that they take up PF4, with an increasing intensity in the fluorescent signal in neural precursor cells (detected by confocal microscopy) after 2h and 6h of incubation, which remained in the cells at 24 h post treatment. Six independent experiments were performed and the results have been added to the manuscript on p. 5 lines 99-105 and included in Supplementary fig. 1a ,b.

Minor:

a) Fig 3a-c; the authors show only the quantification of immunohistochemical experiments, adding the original images would be necessary.

Representative images of Ki67⁺ cells and DCX⁺ cells have been added to the manuscript in Fig. 4 for the 28-day running time point and representative images of DCX⁺ cells have been added in Supplementary fig. 5 for the short and intermediate running durations.

b) Original images depicting DCX+ cells are beneficial here.

As stated in a), we have now included representative images of DCX⁺ cells.

Reviewer #3 (Remarks to the Author):

These data show that the cognition-enhancing effects of endurance training of aged mice are mediated by platelets and can be replicated by the systemic administration of platelet-released exerkine PF4 in a hippocampal neurogenesis-dependent manner. This research approach is very interesting because cognition-enhancing effects may be achieved by platelets, which are very high concentrated in the blood circulation. Especially because it has been described that the platelet secretome contains many regeneration-promoting growth factors.

However, the physiological mode of action of how PF-4 is supposed to induce both its release from platelets and direct signal transduction of neurogenesis in hippocampus remains largely unclear in this work. At this point, it would be also important to study how PF4 would mediate this neurogenesis-promoting effect in a neuronal cell culture model.

We agree that our work does not reveal the precise molecular mechanisms through which exercise activates platelets and thereby causes the release of PF4 into the blood. However, this is beyond the scope of the present study. We do agree with the reviewer that more focus should be given towards understanding the mechanism by which PF4 mediates hippocampal neurogenesis. To address this, we have now performed several additional experiments:

1) We first examined whether PF4 can directly be taken up by adult neural precursor cells, using an established adherent hippocampal monolayer culture model (Bernas, Leiter et al., 2017, *Bio-protocol*). Recombinant PF4 protein was labelled with a fluorescent dye using Lightning Link[®] technology. This enabled specific labelling of the protein and included a quenching step to eliminate any remaining free label. Culturing of the neural precursor cells in the presence of labelled PF4 revealed that the cells take up PF4 in a time-dependent manner, with an increasing intensity in the fluorescent signal in neural precursor cells being detected by confocal microscopy after 2h, 6h and 24 h. These data can be found on p. 5, lines 99-105 and Supplementary fig. 1.

2) To further elucidate the mechanism by which PF4 mediates neurogenesis we performed additional experiments on adherent hippocampal monolayer cells, to determine the effects of PF4 on proliferation (using BrdU labelling; n = 6 individual experiments), cell cycle (using

clickIT[®] EdU technology; n = 7 individual experiments) and neuronal lineage differentiation (using the differentiation assay; n = 6 individual experiments). For the proliferation assay, we cultured adherent hippocampal precursor cells in the presence of PF4 and labelled the cells with the thymidine analogue BrdU. We observed no difference in the number of BrdU-labelled cells between PF4-supplemented and control cultures, suggesting that PF4 had no effect on the proliferative capacity of these cells (Supplementary fig. 1c, d). In support of this we also found no difference in the proportion of PF4-treated cells in the S and G2/M phases of the cell cycle (Supplementary fig. 1e, f). However, we did observe that a higher proportion of cells entered the G1/G0 phases after PF4 treatment (Supplementary fig. 1f), indicating that they had left their proliferative phase and started to differentiate into neurons. To confirm whether PF4 could stimulate neuronal differentiation, we performed a differentiation assay on the adherent cultures and quantified the number of cells that had become β -III-tubulin⁺ neurons and glial fibrillary acidic protein (GFAP)⁺ astrocytes (Supplementary fig. 1g, h). In support of our previous data generated from differentiated neurosphere cultures (Leiter et al., 2019, *Stem Cell Reports*), we found a significant increase in β -III-tubulin⁺ cells in the PF4-treated adherent cultures and no effect on astrocyte differentiation. Importantly, these new data also confirmed our *in vivo* observations, which show that PF4 affects neuronal differentiation without affecting proliferation, at least in the highly proliferative conditions of young animals. A paragraph summarising these data has been included in the revised manuscript on p. 5-6 lines 105-122, and Supplementary fig. 1.

Instead, platelet depletion experiments and platelet proteomics analyzes using TMT shotgun technology were carried out in various mouse models, which are of lesser importance in clarifying this PF-4 neurogenesis-promoting mechanism of action. The authors describe, as if it would be new research findings, that increased platelet activation occurs shortly after exercise. It has long been known that platelets are activated by exercise and during this process release numerous proteins known for platelet activation, such as e.g. B. sCD62P.

We understand that it is already known that exercise activates platelets and that this causes the release of several molecules into the blood. The novel aspect of our study is in that exercise-mediated platelet function promotes adult hippocampal neurogenesis and enhances cognition in ageing. This is particularly highlighted by our platelet depletion experiment, in which aged

exercising mice that are depleted of platelets do not show increases in exercise-induced adult hippocampal neurogenesis and cognition. This is in accordance with our data from exercising PF4 knockout animals, in which exercise-induced increases in proliferating neural precursor cells are also absent. To highlight this novel mechanism, we have revised our discussion to now read (p. 18, lines 438-440):

“Our results provide the first evidence that platelets release exerkinins, including PF4, that increase adult neurogenesis and rejuvenate cognitive function, cementing their important role in the regulation of brain function.”

To address the neurogenesis-promoting aspect of PF4 and its effects on neural precursor cells, we have now performed RNA sequencing on a flow cytometry-purified population of primary dentate gyrus-derived neural precursor cells treated with either PF4 or saline. This analysis revealed several transcriptomic changes in the precursor cells following PF4 treatment, suggesting direct effects on the cells, with a gene ontology (GO) enrichment analysis revealing an enrichment of several upregulated genes in GO categories linked to neuronal differentiation. A paragraph describing this work has been included in the manuscript on p. 7-8, lines 150-184, as inserted below. Corresponding data are displayed in Fig. 2, Supplementary fig. 3, as well as Supplementary tables 1-3).

“To gain further insight into the mechanism by which PF4 affects adult hippocampal neural precursor cells, we performed ribonucleic acid (RNA) sequencing. To do this, adult primary dentate gyrus-derived neural precursors were treated with either PF4 or saline, then isolated by flow cytometry using a biotinylated epidermal growth factor (EGF) complexed to a fluorescent marker (Fig. 2a and Supplementary fig. 3a). RNA sequencing of six samples from each cell population revealed several changes in the transcriptomic signature of EGF⁺ adult neural precursor cells 2 h after PF4 treatment, including 270 genes that were significantly upregulated and 386 genes which were significantly downregulated (Fig. 2b, c, Supplementary fig. 3c and Supplementary table 1). PF4 induced fewer changes in other dentate gyrus cells (EGF⁻ cell population), in which our analysis revealed 110 differentially expressed genes (73 genes upregulated and 37 downregulated; Fig. 2b, Supplementary fig. 3c, d and Supplementary table 2). A comparison of the significantly upregulated and significantly downregulated genes in both cell populations (EGF⁺ and EGF⁻ cells) revealed only 9 and 5 overlapping genes, respectively (Fig. 2d), suggesting differential effects of PF4

treatment in different cell populations. We next performed a gene ontology (GO) enrichment analysis of the differentially expressed genes induced by PF4 in the EGF⁺ neural precursor cell population. This revealed 96 biological processes that were significantly enriched ($p < 0.05$) amongst the upregulated genes and 39 biological processes amongst the genes that were downregulated following PF4 treatment (Fig 2e and Supplementary table 3). In accordance with our data showing a pro-neurogenic effect of PF4, the upregulated genes revealed an enrichment in several GO categories involved in neuronal differentiation, including “cell differentiation” (GO:0030154; $p = 2.1 \times 10^{-5}$), “generation of neurons” (GO:0048699; $p = 2.8 \times 10^{-4}$) “neuron differentiation” (GO:0030182; $p = 0.002$) and “neuron projection development” (GO:0031175; $p = 0.01$) (Supplementary table 3). When viewed as a STRING protein-protein interaction network¹⁶, many of the genes which showed enrichment in GO biological processes related to differentiation also clustered together, suggesting a functional relevance and similarity of these genes (Fig. 2f). Notably, Markov clustering of the genes that were upregulated in EGF⁺ cells following PF4 supplementation also revealed a small STRING network of genes which showed functional enrichment in the GO categories “long-term memory” (GO:0007616; $p = 0.005$), “learning or memory” (GO:0007611, $p = 0.005$) and “positive regulation of synaptic transmission” (GO:0050806, $p = 0.007$), suggesting that PF4 increases the expression of genes associated with learning and memory and synaptic transmission in adult neural precursor cells (Fig. 2g). A GO enrichment analysis of genes that were changed in the EGF⁻ population returned no results, suggesting that PF4 has specific effects on proliferative EGF⁺ adult neural precursor cells.”

Although this work also describes that more weekly endurance training and the persistently recurring platelet activation and associated increased release of PF4 should have a cognitive-promoting effect. In fact, it has been described that endurance exercise, which counteracts vascular disease (PMID: 26557653) as well as Alzheimer's disease, reduces basal platelet activation, platelet reactivity (also PF-4 release should be reduced in circulation) and therefore also has an antithrombotic effect. All of these previous research data in the literature were insufficiently discussed in this work.

We are aware that it has been shown that prolonged exercise reduces the reactivity of platelets. These effects were observed between 8 and 12 weeks following regular endurance training in

humans (Wang et al., 1995, *Arteriosclerosis, Thrombosis, and Vascular Biology* and Wang et al., 1997, *Journal of Applied Physiology*). In our study in mice, we investigate platelet activation after exercise periods which do not exceed 5 weeks. It is difficult to compare these studies, due to the differences in species and the types and durations of exercise. However, it is of note that our longitudinal studies show well-defined peaks of platelet activation after 4 days (Leiter et al., 2019, *Stem Cell Reports*) and 28 days of exercise in young and aged exercising mice, respectively. Following these peaks, platelet activation returned to baseline, but we have not addressed platelet activation in prolonged running paradigms, exceeding 3 weeks for young animals, and 5 weeks for aged mice.

We have now added more detail about the links between exercise, cardiovascular events, neurodegenerative conditions and platelet activation to the discussion on p. 21, lines 503-513, which reads:

“Accumulating evidence demonstrates a clear link between an active lifestyle and brain health, with exercise known to ameliorate cognitive decline in ageing and neurodegenerative conditions¹⁻³. Moreover, several human studies report benefits of exercise on the cardiovascular system^{52,53}. Although our study describes the exercise-induced platelet activation response as beneficial, it is worth noting that several neurodegenerative conditions, as well as cardiovascular events, are linked to chronic platelet activation responses^{54,55}. Interestingly, regular endurance training in humans has been shown to reduce basal levels of platelet reactivity, including their adhesiveness and aggregability, suggesting that chronic exercise has protective effects against cardiovascular disease⁵⁶. In this study we have characterized a platelet activation response to exercise in young and aged mice which appears to be different from other classical platelet activation events, and have identified PF4, which is released from exercise-induced activated platelets, as an anti-geronic exerkine that rejuvenates neurogenesis and cognition in the aged brain.”

Why these effects cannot not also be mediated for example by BDNF for example?

We have now included a paragraph addressing this issue in the discussion on p. 18 lines 433-436, which reads:

“Brain-derived neurotrophic factor (BDNF) is also known to be involved in the exercise-mediated increase in adult hippocampal neurogenesis³³. However, mouse platelets do not carry BDNF³⁴ and it is therefore unlikely that this factor regulates the platelet-mediated neurogenic response to exercise that we observed in mice. This is in line with proteomics analyses performed by ourselves and others³⁵, which found no detectable BDNF protein in the platelets of young or aged mice.”

If they hypothesize that PF4 promotes neurogenesis, why did the researchers perform platelet proteomics?

The proteomic analysis was performed to gain further insight into the molecular changes occurring in platelets (following exercise), as it is likely that PF4 is not the only factor that is changed in platelets capable of inducing neurogenic changes. However, we agree with the reviewer that it is also important to examine the effects of PF4 on the neural stem cells and have therefore now performed transcriptional analyses of a flow cytometry-isolated population of primary EGF⁺ adult neural precursor cells, as described above.

Interestingly, the platelet proteome studies showed that not more PF4 level could be detected in mice after doing endurance training. However, this PF4 data would be important to be shown in the main figures! At this point, it would also have been more important to investigate whether more PF-4 could be detected in the plasma of the exercising mice. Or if the PF4-release is changed in platelets of trained mice.

Given that PF4 is released into the blood by activated platelets, it is not unusual that PF4 levels in the platelets themselves are not significantly higher after exercise. However, although not statistically significant, we do see increased levels of PF4 in the platelets of young mice following 4 days of exercise (2786 mean A.U. in standard-housed mice vs. 3969 mean A.U. in running mice; fold change: 1.42, $p = 0.05$, $n = 5$ mice per group). Given that platelet activation peaks in our 8-week-old mouse model after 4 days of exercise (Leiter et al., 2019, *Stem Cell Reports*), it is likely that the platelets have already released most of the PF4 content into the blood at this timepoint. This is consistent with our previous work showing a significant increase in plasma PF4 after short periods (1 – 4 days) of exercise in young mice (Leiter et al., 2019, *Stem Cell Reports*). We have now included this finding in the results on p 9-10, lines 216-223.

The platelet proteome data of this work showed that the PF4 levels in the platelet proteome of young and old mice were not changed after endurance training as well as other exercise-dependent exerkinases like cathepsin B and clusterin....

Exerkinases are also released from cells and tissues other than platelets. Examples of exerkinases that are also involved in mediating brain plasticity include the liver-derived protein glycosylphosphatidylinositol (GPI)-specific phospholipase D1 (Horowitz et al., 2020, *Science*), muscle-derived cathepsin B (Moon et al., 2016, *Cell Metabolism*) and the complement cascade inhibitor clusterin, which the authors believed to be released from hepatocytes and cardiomyocytes (De Miguel et al. 2021, *Nature*). It is therefore not surprising that the levels of these molecules are unaffected in our platelet proteomic analysis. We have modified our discussion to highlight this fact (p. 18, lines 422-431), which reads:

“However, we still have only a limited understanding of how the systemic effects of exercise are communicated to the brain. Exerkinases, which are released from multiple cells or organs, are emerging as likely mediators of this response. Examples of exerkinases that are involved in mediating brain plasticity include the liver-derived protein glycosylphosphatidylinositol (GPI)-specific phospholipase D1 (Gpld1)⁶, the myokine cathepsin B⁸ and proteins in the complement and coagulation pathways, including clusterin, which the authors suggested is released from hepatocytes or cardiomyocytes⁷. Although our proteomic analysis revealed that these three exerkinases are also present in the platelets of both young and aged mice, their levels remained unaffected by exercise, further suggesting that multiple sources of exerkinases contribute to exercise-mediated brain rejuvenation.”

It should also be additionally noted as further limitations of this study work that proteomic shotgun analysis (bottom-up proteomics) only measures the levels of canonical proteins and cannot distinguish between functional proteoforms, in contrast to top-down proteomics, which also detect functional proteoform differences can. Inactive and active protein such as cathepsin B. (Regulation of cathepsin B functionality is regulated by proteolytic cleavage of cathepsin B).

We thank the reviewer for pointing this out and we have highlighted this limitation in the methods section on p. 37, lines 903-905, which reads:

“It is of note that while this traditional approach identifies a large number of differentially expressed proteins, information about the structural changes is not detected.”

The researchers also found that tropomyosin 1 levels increased after exercise in both old and young mice. Interestingly, it has previously been found that tropomyosin is elevated both in platelets from female Alzheimer's patients and in brains from Alzheimer's cases. But here it is important to know that tropomyosin 1 in platelets has numerous proteoforms and only its high molecular weight protein species are increased in platelets of Alzheimer's patients

This is an interesting point, and one which would be worthy of further investigation in a future study. It is however, beyond the scope of the current study, in which we have chosen to focus on the effects of exercise-released PF4 on adult hippocampal neurogenesis and age-related cognitive decline.

If the researchers want to use their platelet proteome data, it can be helpful to correlate neurogenesis data with the training level of the young and old mice to find potentially significant and hopefully knowledge-enhancing correlations. These results could indicate which protein in the platelet proteome might be significantly involved in neurogenesis.

We thank the reviewer for this interesting idea. We have previously identified a positive correlation between exercise level (running distance) and neural precursor cell proliferation in young C57BL/6J mice (see Overall et al., 2013, *PLoS One*). We have also shown a significant correlation between the number of DCX⁺ cells and spatial learning performance in aged mice following exercise (Blackmore et al., 2021, *iScience*). However, at this point, with a sample size of n = 5-7 mice per group for the proteomics analysis (running or standard-housed), we would not be able to produce reliable correlations between exercise or neurogenesis data with the platelet protein data. This could, however, be implemented in future studies.

REVIEWERS' COMMENTS

Reviewer #1 (Remarks to the Author):

The revised manuscript has addressed my concerns sufficiently. The paper is now substantially improved.

Reviewer #2 (Remarks to the Author):

All my concerns have been addressed. I have no further remarks.

Reviewer #3 (Remarks to the Author):

The authors have revised and answered all questions satisfactorily.